# Attacking the Spike: On the Security of Spiking Neural Networks to Adversarial Examples

## Abstract

Spiking neural networks (SNNs) have attracted much attention for their high energy efficiency and for recent advances in their classification performance. However, unlike traditional deep learning approaches, the analysis and study of the robustness of SNNs to adversarial examples remain relatively underdeveloped. In this work, we focus on advancing the adversarial attack side of SNNs and make three major contributions. First, we show that successful white-box adversarial attacks on SNNs are highly dependent on the underlying surrogate gradient estimation technique, even in the case of adversarially trained SNNs. Second, using the best single surrogate gradient estimation technique, we analyze the transferability of adversarial attacks on SNNs and other state-of-the-art architectures like Vision Transformers (ViTs), as well as CNNs. Our analyzes reveal two key areas where SNN adversarial attacks can be enhanced: no white-box attack effectively exploits the use of multiple surrogate gradient estimators for SNNs, and no single model attack is effective at generating adversarial examples misclassified by both SNNs and non-SNN models simultaneously.

For our third contribution, we develop a new attack, the Mixed Dynamic Spiking Estimation (MDSE) attack to address these issues. MDSE utilizes a dynamic gradient estimation scheme to fully exploit multiple surrogate gradient estimator functions. In addition, our novel attack generates adversarial examples capable of fooling both SNN and non-SNN models simultaneously. The MDSE attack is as much as 91.4% more effective on SNN/ViT model ensembles and provides a $3\times$ boost in attack effectiveness on adversarially trained SNN ensembles, compared to conventional white-box attacks like Auto-PGD. Our experiments are broad and rigorous, covering three datasets (CIFAR-10, CIFAR-100 and ImageNet) and nineteen classifier models (seven for each CIFAR dataset and five models for ImageNet). We will release a full publicly available code repository for the models and attacks upon publication.

## 1 Introduction

There is an increasing demand to deploy machine intelligence to power-limited devices such as mobile electronics and Internet-of-Things (IoT), however, the computation complexity of deep learning models, coupled with energy consumption has become a challenge Kugele et al. (2020); Shrestha et al. (2022). This motivates a new computing paradigm, bio-inspired energy efficient neuromorphic computing. As the underlying computational model, Spiking Neural Networks (SNNs) have drawn considerable interest Davies et al. (2021). SNNs can provide high energy efficient solutions for resource-limited applications. For example, in Rueckauer et al. (2022) it was reported that an SNN consumed $0.66mJ$, $102\ mJ$ per sample on MNIST and CIFAR-10, while a Deep Neural Network (DNN) consumed $111\ mJ$ and $1035\ mJ$, resulting in $168\times$ and $10\times$ energy reduction, respectively. Emerging SNN techniques such as joint thresholding, leakage, and weight optimization using surrogate gradients have all led to improved performance. Both transfer based Lu & Sengupta (2020); Rathi et al. (2020); Rathi & Roy (2021) SNNs and backpropagation (BP) trained-from-scratch SNNs Shrestha & Orchard (2018); Fang et al. (2020; 2021a;b) achieve similar performance to DNNs, while consuming considerably less energy.

On the other hand, the vulnerability of deep learning models to adversarial examples Goodfellow et al. (2014) is one of the main topics that has received much attention in recent research. An adversarial example is an input that has been manipulated with a small amount of noise such that a human being can correctly classify it. However, the adversarial example is misclassified by a machine learning model with high confidence. A large body of literature has been devoted to the development of both adversarial attacks Tramer et al. (2020) and defenses Zhang et al. (2020) for CNNs.

As SNNs become more accurate and more widely adopted, their security vulnerabilities will emerge as an important issue. Recent work has been done to study some of the security aspects of the SNN El-Allami et al. (2021); Sharmin et al. (2019; 2020); Goodfellow et al. (2015); Kundu et al. (2021); Liang et al. (2021), although not to the same extent as CNNs. A unique challenge arises in the study of SNN due to spiking neuron's non-differentiable binary activation, i.e., neuron's output can only be 1 (fire) or 0 (not fire). As true gradients do not exist in SNNs, surrogate gradient Neftci et al. (2019), which is a technique to estimate the gradient of spikes, has been proposed to enable BP. White-box attacks in adversarial machine learning rely on accurate gradient calculation. However, the choice of surrogate estimator for the gradient calculations in SNNs is highly flexible. How the gradient estimation affects white-box remains unknown. To the best of our knowledge, there have not been rigorous analyses done on how the different choices of gradient estimations can effect white-box SNN attacks. In addition, it is an open question whether SNN adversarial examples are misclassified by other state-of-the-art models like Vision Transformers (ViTs). Finally, there has not been any general attack method developed to break both SNNs and CNNs/ViTs simultaneously. Thus in our paper, we specifically focus on three key security aspects:

1. *How do the use of different SNN gradient estimation functions impact the effectiveness of white-box attacks?*
2. *As SNNs have shown more robustness in previous studies Sharmin et al. (2019); Liang et al. (2021), do adversarial examples generated by SNNs transfer to other models such as Vision Transformers and CNNs and vice versa?*
3. *Can white-box attacks leverage different gradient estimation functions to more effectively attack SNNs? Additionally, can a white-box attack be developed that effectively target both SNNs and CNNs/Vision Transformers, closing the transferability gap and achieving a high success rate?*

**Paper Organization**: These three questions are intrinsically linked and form the outline for our paper. After introducing the types of SNNs in Section 2 we show the choice of gradient estimator plays a major role in the success of SNN white-box attacks in Section 3. Then, using the single best gradient estimator, we analyze the transferability of adversarial examples between SNNs and other SOTA architectures (as posed in our second question) in Section 4. Based on the outcome of the second question (low attack transferability), an important attack issue arises: current SOTA white-box attacks cannot break an ensemble of SNNs and non-SNN models. To solve this issue, we further develop a new attack, the Mixed Dynamic Spiking Estimation (MDSE) attack in Section 5. The advantages of our new attack are two-fold: Through dynamic spiking surrogate gradient estimation we create a more effective SNN specific attack framework. Second, by mixing gradients from multiple models we are able to craft adversarial examples that are misclassified by both non-SNN and SNN models simultaneously, bridging the transferability gap. We empirically demonstrate the superiority of the MDSE attack to MIM Dong et al. (2018), PGD Madry et al. (2018), SAGA Mahmood et al. (2021b) and Auto-PGD Croce & Hein (2020) in Section 6.

**Main Contributions**: Overall, we conduct rigorous analyses and experiments with 19 models across three datasets (CIFAR-10, CIFAR-100, and ImageNet) and four adversarial training methods. We consider two recently proposed SNN-based adversarial training methods: Temporal Information Concentration (TIC) Kim et al. (2023) and HIRE Kundu et al. (2021). Additionally, we explore SNN models trained with techniques originally designed for CNNs, such as Diffusion Model (DM) enhanced adversarial training Wang et al. (2023) and Friendly Adversarial Training (FAT) Zhang et al. (2020). Our surrogate gradient estimator results on normal and adversarially trained SNNs consistently show that an optimal SG is crucial for accurately evaluating the robustness of SNNs. The transferability results highlight the low attack transferability among SNNs and non-SNNs, providing new insights into SNN security. Our newly proposed attack, MDSE, achieves higher attack success rates on SNN/ViT/CNN ensembles, with improvements of up to 91.4%. Additionally, MDSE is three times more effective than conventional white-box attacks like Auto-PGD when targeting

adversarially trained SNN ensembles. These findings significantly advance the security development of SNN adversarial machine learning.

## 2 Spiking Neural Network Types

In this section, we discuss the basics of the SNN architecture and of neural encoding. Widely used Leaky Integrate and Fire (LIF) neuron can be described by a system of difference equations as follows Shrestha et al. (2022):

$$V[t] = \alpha V[t-1] + \sum_i w_i S_i[t] - \vartheta O[t-1] \tag{1a}$$

$$O[t] = u(V[t] - \vartheta) \tag{1b}$$

$$u(x) = 0, x < 0 \text{ otherwise } 1 \tag{1c}$$

where $V[t]$ denotes neuron's membrane potential. $\alpha \in (0,1]$ is a time constant, which controls the decay speed of membrane potential. When $\alpha = 1$, the model becomes Integrate and Fire (IF) neuron. $S_i[t]$ and $w_i$ are $i_{th}$ input and the associated weight. $\vartheta$ is the neuron's threshold, $O[t]$ is the neuron's output function, $u(\cdot)$ is the Heaviside step function. If $V[t]$ exceeds the threshold $\vartheta$, the neuron will fire a spike, hence $O[t]$ will be 1. Then, at the next time step, $V[t]$ will be decreased by $\vartheta$ in a procedure referred to as a reset Shrestha et al. (2022).

Note that, in contrast to the continuous input domains of DNNs, in SNNs information is represented by discrete, binary spike trains. Therefore, data has to be mapped to the spike domain for an SNN to process, which is known as neural encoding Shrestha et al. (2022). A popular way to achieve such a mapping is by using direct encoding Wu et al. (2019); Rathi & Roy (2021). This encoding can reduce inference latency by a factor of $5-100$ Rathi & Roy (2021). Recent works have achieved state-of-the-art results with this coding scheme Rathi & Roy (2021); Fang et al. (2021a); Kundu et al. (2021); Fang et al. (2020). Hence, all experiments in this paper employ direct coding.

### 2.1 Spiking Neural Network Training

The neurons within the SNN have non-differentiable activation functions, which makes directly applying BP challenging Zhang & Li (2020); Tavanaei et al. (2019). Broadly speaking, there are two common techniques for training an SNN, conversion based or surrogate gradient based training. In conversion based training, it is possible to pre-train a DNN model and map the weights to an SNN. However, simply mapping the weights suffers from performance degradation due to non-ideal input-spike rate linearity, over activation, and under activation Rathi & Roy (2021); Diehl et al. (2015). Additional post-processing and fine tuning are required to compensate for the performance degradation such as weight-threshold balancing Diehl et al. (2015).

A second way to train SNNs is through the use of surrogate gradient BP. Equation 1a - 1c reveal that SNNs have a similar form to Recurrent Neural Networks (RNNs). The membrane potential is dependent on input and historical states. Equation 1a is actually differentiable, thereby making it possible to unfold the SNN and use BP to train it.

Following the standard way to unfold the SNN, we can derive the Backpropagation rule. Let $o_i^l[t]$ represents the output of $i_{th}$ neuron in layer $l$ at time $t$, and it derivative with respect to the loss function $\mathcal{L}$ be $\delta_i^l[t] = \frac{\partial \mathcal{L}}{o_i^l[t]}$. By applying the chain rule, $\delta_i^l[t]$ can be calculated recursively Neftci et al. (2019):

$$\delta_i^l[t] = u'(V_i^l[t])(\sum \delta_n^{l+1})W_i^{\mathsf{T},l}k + \delta_i^l[t+1]\vartheta) \tag{2}$$

where $u'(\cdot)$ is the derivative of function $u(\cdot)$, which will be discussed in section 3. Weight can be updated as Neftci et al. (2019):

$$\Delta W_{ij}^l \propto \frac{\partial \mathcal{L}}{W_{ij}^l} = \sum_{t=0}^{T-1} \delta_i^l[t] o_j^{l-1}[t] \tag{3}$$

The challenge is Equation 1c, i.e. the Heaviside step function $u(\cdot)$ is non-differentiable. To overcome this issue, the surrogate gradient method has been proposed Neftci et al. (2019), which allows the Heaviside step function's derivative $u'(\cdot)$ to be approximated by some smooth function. Using a surrogate gradient enables SNN training with BP and achieves comparable performance to DNNs Shrestha & Orchard (2018); Fang et al. (2021a). There are multiple viable choices for the surrogate gradient method.

## 2.2 Spiking Neural Network Adversarial Training

We also consider defenses based on SNNs, in addition to vanilla (undefended) SNN models. One of the most common ways to defend against adversarial attacks is through adversarial training Madry et al. (2018). In our work, we consider two recently proposed SNN-based adversarial training methods: Temporal Information Concentration (TIC) Kim et al. (2023) and HIRE Kundu et al. (2021). Additionally, we explore SNN models trained with techniques originally designed for CNNs, such as Diffusion Model (DM) enhanced adversarial training Wang et al. (2023) and Friendly Adversarial Training (FAT) Zhang et al. (2020).

**Temporal Information Concentration (TIC)** Kim et al. (2023) indicates the information in SNN shifts from latter timesteps to earlier timesteps as training progresses. The defense proposed a loss function to control the Fisher information value at each timestep:

$$L_t(\theta, \alpha) = |L_t(\theta) - \alpha| \tag{4}$$

$$L(\theta, \alpha) = \frac{1}{T} \sum_{t=1}^{T} L_t(\theta, \alpha) \tag{5}$$

We apply Eq. 4 across $T$ timesteps to force the loss function to a value around $\alpha$, ensuring that the Fisher information shows a similar trend for all timesteps. The key takeaway is that the SNN model exhibiting temporal concentration behavior (smaller Fisher trace as time goes on) might have better robustness.

*Why we selected it*: TIC was selected for its potential to specifically address the unique temporal dynamics of SNNs with improved robustness that is directly related to the SNN features.

**HIRE-SNN (Spike Timing Dependent Backpropagation)** Kundu et al. (2021) is a training algorithm designed to enhance the inherent robustness of conversion-based SNNs. It partitions the total time steps $T$ into $N$ equal-length periods. The gradients with respect to the weights $\delta_w$ and perturbations $\kappa$, as well as threshold $v_t$ and leak $l_k$ parameters, are calculated and updated over small intervals of $\lfloor \frac{T}{N} \rfloor$ steps. The HIRE-SNN training process involves calculating gradients and updating weights based on the following steps:

$$\delta_w \leftarrow \mathbb{E}_{(x,y)\in B} \left[ \nabla_w \mathcal{L} \left( g(x + \kappa, y; \frac{T}{N}) \right) \right] \tag{6}$$

$$\delta_x \leftarrow \left[ \nabla_x \mathcal{L} \left( g(x + \kappa, y; \frac{T}{N}) \right) \right] \tag{7}$$

$$\kappa \leftarrow \text{clip}(\kappa + \epsilon_s \cdot \text{sign}(\delta_x), -\epsilon_t, \epsilon_t) \tag{8}$$

$$\mathbf{W} \leftarrow \mathbf{W} - \eta \cdot \delta_w \tag{9}$$

In these equations, $(x, y)$ represents the input data and label, $B$ denotes the batch of data, and $\delta_x$ represents the gradients with respect to inputs. $\mathbf{W}$ represents the model weights. $\epsilon_s$ is the step size for the perturbation, $\epsilon_t$ is the perturbation limit, $\eta$ is the learning rate. This method allows the model to be trained with various adversarial image variants without incurring additional training time.

*Why we selected it*: HIRE was chosen for its ability to enhance SNN robustness against temporal perturbations, offering a strong benchmark for assessing the effectiveness of our proposed adversarial attacks across different SNN models.

**Friendly Adversarial Training (FAT)** Zhang et al. (2020) focuses on identifying the least adversarial data that minimizes the loss among the adversarial data that is misclassified. This training approach employs a modified version of PGD called PGD-K-$\tau$. In PGD-K-$\tau$, $K$ refers to the number of iterations used for PGD,

and $\tau$ is a hyperparameter that allows early stopping in the PGD generation of adversarial examples if the sample is already misclassified. The FAT method involves updating the model parameters $\theta$ as follows:

$$\theta \leftarrow \theta - \eta \frac{1}{m} \sum_{i=1}^{m} \nabla_\theta \ell(f_\theta(\tilde{x}_i), y_i) \tag{10}$$

where $\theta$ represents the model parameters, $\tilde{x}_i$ represents the adversarially perturbed input, $y_i$ is the true label, $\ell$ is the loss function, $\eta$ is the learning rate, and $m$ is the batch size.

*Why we selected it*: FAT is an adversarial training method that can maintain higher clean accuracy due to its early stopping PGD algorithm during training. However, FAT has only been tested with CNN variants and never with SNN models. As FAT is one prominent recent adversarial training algorithm, testing its effectiveness with SNNs is of interest.

**Diffusion Model (DM) Enhanced Adversarial Training** Wang et al. (2023) involves the use of class-conditional elucidating diffusion models (EDM) Karras et al. (2022) to generate augmented datasets for CIFAR-10 and CIFAR-100. These datasets are used in the TRadeoff-inspired Adversarial DEfense via Surrogate-loss minimization (TRADES) Zhang et al. (2019) pipeline, which employs a classification-calibrated loss theory to balance accuracy and robustness. The loss function used in TRADES is:

$$\mathcal{L}_{\text{TRADES}} = \mathcal{L}_{\text{CE}}(f_\theta(x), y) + \beta \cdot \max_{x' \in \mathcal{B}(x, \epsilon)} \mathcal{L}_{\text{KL}}(f_\theta(x), f_\theta(x')) \tag{11}$$

where $\mathcal{L}_{\text{CE}}$ is the cross-entropy loss, $f_\theta(x)$ is the model output for input $x$ with parameters $\theta$, $y$ is the true label, $\beta$ is the hyperparameter to control the trade-off, and $\mathcal{L}_{\text{KL}}$ is the Kullback-Leibler divergence between the outputs of the original input $x$ and the perturbed input $x'$. $\mathcal{B}_p(x, \epsilon) := \{x' \mid \|x' - x\|_p \leq \epsilon\}$ denotes that the input $x'$ is constrained into the $\ell_p$ norm, where $\epsilon$ is the maximum perturbation constraint.

*Why we selected it*: DM was included to assess how data augmentation and loss optimization techniques originally developed for CNNs perform in enhancing SNN robustness against adverarial attacks. In addition, DM provides SOTA robustness results on CIFAR-10 with CNN architectures, making it an ideal candidate to implement and test in the SNN domain.

**The Importance of Adversarial Training**: To the best of our knowledge, we are the first to implement DM and FAT adversarial training techniques on SNNs. We are also the first to compare existing SNN-specific adversarial training (TIC and HIRE) to DM and FAT. While this alone is not a major contribution, in the context of developing adversarial attacks, it is critical to include these types of analyses. This is because existing adversarial attacks can readily be adapted to new undefended architectures Mahmood et al. (2021b) yielding a high attack success rate. However, on defended models or model ensembles, existing adversarial attacks may not be effective. Experimenting with adversarial training defense methods are key for accurately assessing the robustness of SNNs to SOTA adversarial attacks. Many other adversarial training algorithms exist and are being proposed or developed recently Ozdenizci & Legenstein; Liu et al.. While we cannot test all of them, we do include Liu et al. for discussion and comparison in Section 6.

## 3  Surrogate Gradient Estimation

In both neural network training and white-box adversarial machine learning attacks, the fundamental computation requires backpropagating through the model. Due to the non-differentiable structure of SNNs Neftci et al. (2019), this requires using a surrogate gradient estimator. In Zenke & Vogels (2021), it was shown that gradient based SNN training was robust to different derivative shapes. In Wu et al. (2019), it was demonstrated that there are multiple different gradient estimators that can lead to reasonably good performance on MNIST, N-MNIST and CIFAR-10.

While there exist multiple viable surrogate gradient estimators for SNN training, in the field of adversarial machine learning, precise gradient calculations are paramount. Incorrect gradient estimation on models leads to a phenomenon known as gradient masking Athalye et al. (2018a). Models that suffer from gradient masking appear robust, but only because the model gradient is incorrectly calculated in white-box attacks performed

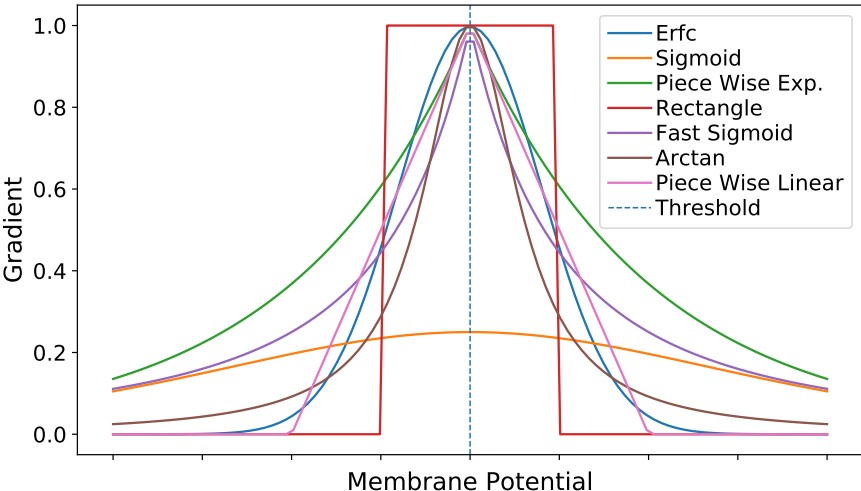

Figure 1: Different surrogate gradient functions.

against them. This issue has led to many published models and defenses to claim security, only to later be broken when correct gradient estimators were implemented Tramer et al. (2020). To the best of our knowledge, this issue has not been thoroughly explored for SNNs in the context of adversarial examples. Hence, we run white-box attacks on SNNs using different surrogate gradient estimators, to empirically understand their effect on attack success rate. In our analyses, we experiment with undefended SNNs and four types of adversarial trained SNN models.

### 3.1 Representative Surrogate Gradient Functions

Here we give a brief introduction to some representative surrogate gradient functions. We denote the derivative of Heaviside Step Function $u(x)$ as $u'(x)$, threshold as $\vartheta$. The surrogate gradients investigated in this work are discussed as follows, and their shapes are shown in Figure 1.

**Sigmoid** Bengio et al. (2013) is a pioneer work which studies gradient estimation of non-smooth neuron. It indicates that a hard threshold function's derivative can be approximated by that of a Sigmoid function. Such that $u'(x)$ can be approximated as:

$$u'(x) \approx \frac{e^{\vartheta-x}}{(1+e^{\vartheta-x})^2} \tag{12}$$

**Erfc** Fang et al. (2020) takes a bio-inspired approach, it proposes to use the Poisson neuron's spike rate function, which can be characterized by a complementary error function (erfc). Its derivative is given in equation 13, where $\sigma$ controls the sharpness.

$$u'(x) \approx \frac{e^{-\frac{(\vartheta-x)^2}{2\sigma^2}}}{\sqrt{2\pi}\sigma} \tag{13}$$

**Arctan** Fang et al. (2021b;a) used Arctangent function as surrogate gradient, achieving state-of-the-art results on various datasets. The surrogate gradient is given by:

$$u'(x) \approx \frac{1}{1+\pi^2(x-\vartheta)^2} \tag{14}$$

**Piece-wise linear function (PWL)** Neftci et al. (2019) is the first work that formally established the framework of Surrogate Gradient method. It studied PWL function as gradient surrogate. In addition, PWL

Table 1: APGD attack success rate for transfer SNN VGG-16 model on CIFAR-10 and CIFAR-100 with respect to different surrogate gradients.

| | CIFAR-10 | | | | | CIFAR-100 | | | | |
|---|---|---|---|---|---|---|---|---|---|---|
| | $\epsilon$ | | | | | $\epsilon$ | | | | |
| | 0.0062 | 0.0124 | 0.0186 | 0.0248 | 0.031 | 0.0062 | 0.0124 | 0.0186 | 0.0248 | 0.031 |
| PWL | 36.9% | 63.2% | 80.0% | 88.1% | 93.5% | 76.1% | 92.1% | 96.7% | 98.6% | 99.1% |
| Erfc | 37.1% | 63.3% | 80.0% | 88.3% | 93.1% | 76.2% | 91.8% | 96.8% | 98.6% | 99.2% |
| Sigmoid | 5.6% | 15.8% | 25.1% | 33.6% | 41.7% | 37.7% | 60.1% | 72.2% | 81.5% | 85.4% |
| Piecewise Exp. | 3.2% | 6.1% | 10.2% | 15.8% | 21.8% | 10.1% | 19.8% | 31.4% | 38.5% | 46.8% |
| Rectangle | 35.3% | 60.4% | 75.2% | 84.7% | 90.2% | 73.3% | 90.2% | 95.1% | 97.9% | 98.6% |
| Fast Sigmoid | 24.2% | 45.8% | 62.0% | 76.0% | 83.5% | 68.2% | 88.2% | 94.9% | 97.9% | 99.1% |
| Arctan | 35.5% | 60.5% | 79.1% | 88.4% | 94.1% | 76.3% | 92.2% | 97.1% | 98.6% | 99.3% |

is also used in Rathi & Roy (2021); Bellec et al. (2018). Its formulation is given by:

$$u'(x) \approx \max(0, \vartheta - |x|) \tag{15}$$

**Fast Sigmoid** Zenke & Ganguli (2018) uses Fast Sigmoid as a replacement of the Sigmoid function, the purpose is to avoid expensive exponential operation and to speed up computation. It is defined as:

$$u'(x) \approx \frac{1}{1 + (1 + |x - \vartheta|)^2} \tag{16}$$

**Piece-wise Exponential** Shrestha & Orchard (2018) suggests that Probability Density Function (PDF) for a spiking neuron to change its state (fire or not) can approximate the derivative of the spike function. Spike Escape Rate, which is a piece-wise exponential function, can be a good candidate to characterize this probability density. It is given by equation 17, where $\alpha$ and $\beta$ are two hyperparamaters.

$$u'(x) \approx \frac{1}{\alpha e^{-\beta |x - \vartheta|}} \tag{17}$$

**Rectangular function** is used by Wu et al. (2018; 2019), which are two representative works that empirically demonstrated that Surrogate Gradient together with Backpropagation Through Time can be used to train high performance SNNs. It is given by equation 18, where $\alpha$ is a hyperparameter that controls height and width.

$$u'(x) \approx \frac{1}{\alpha} \text{sign}(|v - \vartheta| < \frac{\alpha}{2}) \tag{18}$$

### 3.2 Surrogate Gradient Estimator Experiments

**Experimental Setup:** We evaluate the attack success rate of aforementioned gradient estimators on SNNs trained with and without adversarial training. For the attack, we use one of the most common white-box attacks, the Auto Projected Gradient Descent (Auto-PGD) attack Croce & Hein (2020) with respect to the $l_\infty$ norm. When conducting Auto-PGD, we keep the model's forward pass unchanged, and the surrogate gradient function is substituted in the backward pass only. For the undefended (vanilla) SNNs we test 3 types of SNNs on CIFAR-10/100 Krizhevsky et al. (2009) and 2 types of SNNs on ImageNet Krizhevsky et al. (2012) using 7 different surrogate gradient estimators. We test the Transfer SNN VGG-16 Rathi & Roy (2021), the BP SNN VGG-16 Fang et al. (2020), a Spiking Element Wise (SEW) ResNet Fang et al. (2021a), and Vanilla Spiking ResNet Zheng et al. (2021).

**Vanilla SNN Experimental Analysis:** The results of our surrogate gradient estimation experiments are shown in Figure 2. For each model and each gradient estimator, we vary the maximum perturbation bounds from $\epsilon$=0.0062 to $\epsilon$=0.031 on the x-axis and run the Auto-PGD attack on 1000 (CIFAR-10 and CIFAR-100),

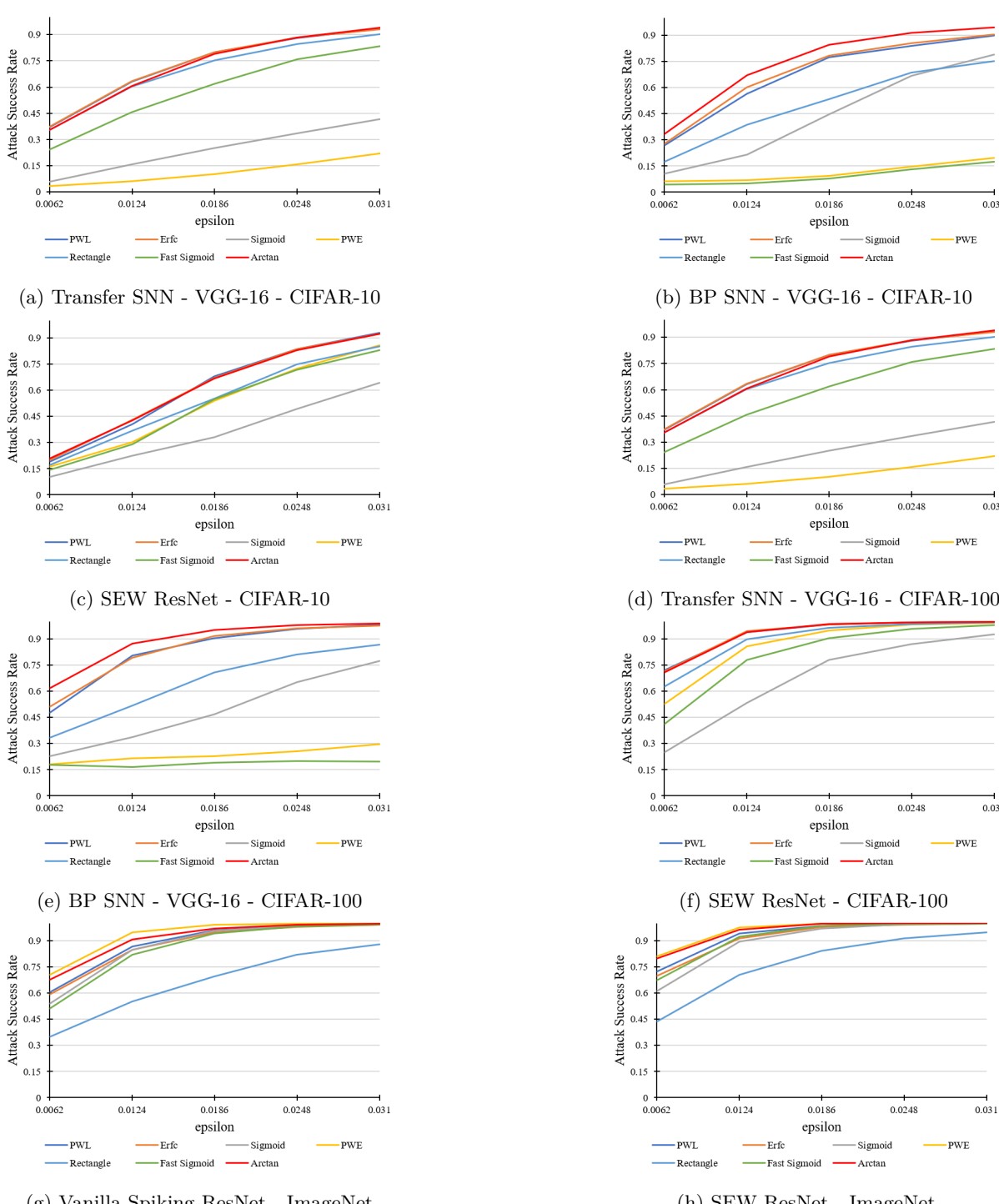

(a) Transfer SNN - VGG-16 - CIFAR-10

(b) BP SNN - VGG-16 - CIFAR-10

(c) SEW ResNet - CIFAR-10

(d) Transfer SNN - VGG-16 - CIFAR-100

(e) BP SNN - VGG-16 - CIFAR-100

(f) SEW ResNet - CIFAR-100

(g) Vanilla Spiking ResNet - ImageNet

(h) SEW ResNet - ImageNet

Figure 2: White-box attack on SNN models using different surrogate gradients for CIFAR-10, CIFAR-100 and ImageNet. Every curve corresponds to the performance of an attack with a specific surrogate gradient. The y-axis is accuracy, the x-axis is epsilon. For CIFAR-10/100, arctan produces the highest attack success rate. On ImageNet models, PWE performs best. Numerical values of the results are given in Table 1 (Transfer SNN), Table 2 (BP SNN), Table 3 (SEW ResNet), Table 4 (Vanilla Spiking ResNet) respectively.

and 2000 (ImageNet) clean, correctly identified and class-wise balanced samples from the validation set. The corresponding robust accuracy is then measured on the y-axis. Our results show that unlike what the literature reported for SNN training Wu et al. (2019), the choice of surrogate gradient estimator hugely

Table 2: APGD attack success rate for BP SNN VGG-16 model on CIFAR-10 and CIFAR-100 with respect to different surrogate gradients.

| | CIFAR-10 | | | | | CIFAR-100 | | | | |
|---|---|---|---|---|---|---|---|---|---|---|
| | $\epsilon$ | | | | | $\epsilon$ | | | | |
| | 0.0062 | 0.0124 | 0.0186 | 0.0248 | 0.031 | 0.0062 | 0.0124 | 0.0186 | 0.0248 | 0.031 |
| PWL | 26.7% | 56.5% | 77.3% | 83.9% | 89.9% | 47.6% | 80.4% | 90.3% | 95.7% | 98.2% |
| Erfc | 27.6% | 60.1% | 78.4% | 85.6% | 90.6% | 51.0% | 79.2% | 91.7% | 96.1% | 97.7% |
| Sigmoid | 10.6% | 21.5% | 44.7% | 66.8% | 78.8% | 22.9% | 33.7% | 46.9% | 65.3% | 77.4% |
| Piecewise Exp. | 6.1% | 6.8% | 9.5% | 14.7% | 19.6% | 18.0% | 21.4% | 22.7% | 25.5% | 29.5% |
| Rectangle | 17.5% | 38.6% | 53.3% | 68.5% | 75.1% | 33.3% | 51.7% | 70.9% | 81.1% | 86.8% |
| Fast Sigmoid | 4.4% | 5.1% | 7.9% | 13.1% | 17.4% | 17.8% | 16.6% | 18.9% | 19.8% | 19.7% |
| Arctan | 33.3% | 67.1% | 84.5% | 91.3% | 94.6% | 61.8% | 87.4% | 95.1% | 98.0% | 99.0% |

Table 3: APGD attack success rate for SEW ResNet 18 model on CIFAR-10, CIFAR-100 and ImageNet with respect to different surrogate gradients.

| | CIFAR-10 | | | | | CIFAR-100 | | | | | ImageNet | | | | |
|---|---|---|---|---|---|---|---|---|---|---|---|---|---|---|---|
| | $\epsilon$ | | | | | $\epsilon$ | | | | | $\epsilon$ | | | | |
| Surrogate Grad. | 0.0062 | 0.0124 | 0.0186 | 0.0248 | 0.031 | 0.0062 | 0.0124 | 0.0186 | 0.0248 | 0.031 | 0.0062 | 0.0124 | 0.0186 | 0.0248 | 0.031 |
| PWL | 18.9% | 40.6% | 67.9% | 83.6% | 93.0% | 72.0% | 93.9% | 98.2% | 99.4% | 99.8% | 72.5% | 94.1% | 98.6% | 99.6% | 99.9% |
| Erfc | 19.9% | 43.0% | 67.2% | 83.7% | 92.2% | 71.5% | 94.4% | 98.4% | 99.5% | 99.8% | 69.9% | 91.4% | 98.0% | 99.1% | 99.8% |
| Sigmoid | 10.2% | 22.6% | 33.1% | 49.3% | 64.3% | 25.0% | 53.2% | 78.0% | 86.9% | 92.5% | 61.2% | 89.7% | 97.1% | 99.4% | 99.9% |
| Piecewise Exp. | 16.1% | 30.2% | 53.8% | 72.5% | 85.9% | 52.8% | 85.8% | 94.7% | 98.2% | 99.1% | 81.2% | 97.8% | 99.8% | 100.0% | 100.0% |
| Rectangle | 17.3% | 36.8% | 55.1% | 74.9% | 85.1% | 62.8% | 89.7% | 96.5% | 98.5% | 99.4% | 43.8% | 70.5% | 84.2% | 91.5% | 94.8% |
| Fast Sigmoid | 14.4% | 28.9% | 54.8% | 71.6% | 83.0% | 41.3% | 77.9% | 90.4% | 95.6% | 97.9% | 67.4% | 92.4% | 98.7% | 99.7% | 99.9% |
| Arctan | 20.9% | 42.8% | 66.8% | 83.1% | 92.3% | 70.8% | 94.0% | 98.6% | 99.4% | 99.8% | 79.7% | 96.4% | 99.7% | 100.0% | 100.0% |

impacts SNN attack performance. In most cases, the arctan yields the lowest accuracy (the highest attack success rate), which includes Transfer SNN C10, BP SNN C10, Transfer SNN C100, BP SNN C100, SEW ResNet C100 experiments.

This trend does not occur for ImageNet, where PWE performs best and arctan performs second best in Vanilla Spiking ResNet and ImageNet experiment. And in SEW ResNet ImafeNet experiment, both PWE and and arctan achieve 100% attack successful rate.

The worst gradient estimator varies in different experiments. For example, though PEW achieves best ASR in two experiment on ImageNet dataset, it has lowest ASR in Transfer SNN C10, BP SNN C10, Transfer SNN C100, BP SNN C100. And sigmoid performs worst in SEW SNN 10 and SEW SNN C100.

Results also show that there is significan performance gap between the best and worst gradient estimator. For example, in BP SNN C10, the arctan achieves 64.6% ASR, while ASR of PWE is merely 19.6%; and in Transfer SNN C100, arctan achieves 99.3% ASR, however PWE only achieves 46.8% ASR.

To reiterate, this set of experiments highlights a significant finding: *for SNNs, choosing the right surrogate gradient estimator is critical for achieving a high white-box attack success rate.*

## 3.3 Adversarial Trained SNN Experimental Analysis

To further validate the substantial influence of the surrogate gradient estimator (SG), we consolidate four state-of-the-art adversarial training (AT) methods and conduct training on SNNs in our study. Specifically, we modify two effective adversarial training methods from the CNN domain, namely DM Wang et al. (2023) and FAT Zhang et al. (2020), for SNN training. Additionally, we introduce two newly proposed adversarial training methods for SNNs, denoted as HIRE Kundu et al. (2021) and TIC Kim et al. (2023). We adopt these AT methods for SNNs and perform MIM, PGD, and Auto-PGD attacks on the trained SNNs using different surrogate gradient estimators. We set the maximum perturbation bounds $\epsilon = 0.031$ and attack steps

Table 4: APGD attack success rate for Vanilla Spiking ResNet 18 model on ImageNet with respect to different surrogate gradients.

| ImageNet | | | | | |
|---|---|---|---|---|---|
| | $\epsilon$ | | | | |
| Surrogate Grad. | 0.0062 | 0.0124 | 0.0186 | 0.0248 | 0.031 |
| PWL | 60.4% | 86.7% | 96.1% | 98.6% | 99.5% |
| Erfc | 59.4% | 85.0% | 94.9% | 98.0% | 99.1% |
| Sigmoid | 54.0% | 84.9% | 95.7% | 98.5% | 99.8% |
| Piecewise Exp. | 70.4% | 94.7% | 99.1% | 99.9% | 100.0% |
| Rectangle | 35.1% | 55.2% | 69.5% | 82.1% | 87.9% |
| Fast Sigmoid | 51.2% | 82.1% | 94.3% | 98.3% | 99.3% |
| Arctan | 67.8% | 90.7% | 97.0% | 99.2% | 99.7% |

Table 5: White box attack success rate for ResNet-18 SNN model with DM adversarial training method on CIFAR-10, CIFAR-100 with respect to different surrogate gradients.

| CIFAR-10 | | | | | | | |
|---|---|---|---|---|---|---|---|
| | Arctan | PWL | Erfc | Sigmoid | PWE | Rectangle | Fast Sigmoid |
| MIM | 37.6% | 37.0% | 36.9% | 39.0% | 21.9% | **41.0%** | 18.5% |
| PGD | 38.0% | 35.9% | 37.4% | 37.0% | 23.2% | **38.7%** | 18.3% |
| Auto-PGD | 55.4% | 54.5% | 54.4% | 55.5% | 40.4% | **56.0%** | 34.1% |
| CIFAR-100 | | | | | | | |
| | Arctan | PWL | Erfc | Sigmoid | PWE | Rectangle | Fast Sigmoid |
| MIM | 46.6% | 44.9% | 47.1% | 47.5% | 35.9% | **48.9%** | 29.7% |
| PGD | **49.6%** | 43.9% | 46.4% | 46.3% | 36.8% | 47.5% | 30.9% |
| Auto-PGD | **64.3%** | 61.7% | 63.1% | 63.4% | 53.0% | 64.0% | 44.8% |

to 40 for all three attacks, with a step size $\epsilon_{step} = 0.01$ for MIM and PGD. We run 1000 clean, correctly identified, and class-wise balanced samples from the validation set on CIFAR-10 and CIFAR-100.

The DM adversarial trained SEW ResNet18 SNNs utilized TRADES5 with 10M and 1M augment data as per the original paper settings on CIFAR-10 and CIFAR-100. Notably, this adversarial training yields the highest robustness among all investigated methods but achieves lowest clean model accuracy (66.8% for CIFAR-10 and 41.0% for CIFAR-100). The attack results are shown in Table 5. For FAT training, the implementation employs early-stopped PGD for ease of adaptation. We maintain consistency with the original paper's approach by employing PGD-10-5 ($k = 10$, and $\tau = 5$) to train SEW ResNet18 SNNs on CIFAR-10 (73.2%) and CIFAR-100 (40.8%). While the trained SNNs demonstrate some level of robustness, it is not as robust as results shown for CNNs as presented in Wang et al. (2023), especially for Auto-PGD results on CIFAR-100 SNN. This discrepancy could be because the FAT training method is originally designed for CNNs and may not be fully adapted to the SNN settings. The attack results are detailed in Table 6.

To train the SNNs on HIRE SNNs, we follows the original paper's methodology, dividing time steps into two equal-length intervals and introducing input noise after each period during training. For CIFAR-10 and CIFAR-100 datasets, we trained VGG-16 (89.0%) and VGG-11 (66.1%), respectively. Although the SNNs achieve higher accuracy, they demonstrate very low robustness when the correct surrogate gradient estimator is chosen, as shown in Table 7. As for SNNs with the TIC method, we follow the guidance provided in the paper and train ResNet-19 SNNs with $\alpha = 1e-3$ for CIFAR-10 and $\alpha = 1e-4$ for CIFAR-100. Although the SNNs achieve high accuracy (92.3% and 72.1%), as stated in the paper, their robustness is not strong, even when different estimators are used, as indicated in Table 8.

**Surrogate Estimator Discussion**: We summarize the attack success rate using the best and worst possible Surrogate Gradient Estimator (SG) for Auto-PGD with $\epsilon = 0.031$ on CIFAR-10/100. It can clearly be seen from Figure. 3 that the choice of estimator is extremely significant in how effective the attack is. If the worst estimator was used, the attack success rate would be significantly lower than if the best estimator was used.

Table 6: White box attack success rate for ResNet-18 SNN model with FAT adversarial training method on CIFAR-10, CIFAR-100 with respect to different surrogate gradients.

| | Arctan | PWL | Erfc | Sigmoid | PWE | Rectangle | Fast Sigmoid |
|---|---|---|---|---|---|---|---|
| | CIFAR-10 | | | | | | |
| MIM | 46.4% | 45.9% | 45.9% | 46.8% | 28.5% | **49.3%** | 25.2% |
| PGD | **47.9%** | 45.9% | 47.1% | 46.8% | 27.7% | 45.9% | 25.6% |
| Auto-PGD | 73.0% | 71.5% | 72.3% | **73.2%** | 54.6% | 69.7% | 55.1% |
| | CIFAR-100 | | | | | | |
| MIM | 69.8% | 70.7% | 70.4% | **71.0%** | 55.6% | 70.5% | 50.4% |
| PGD | 69.2% | 69.6% | **71.1%** | 69.2% | 54.3% | 69.1% | 49.4% |
| Auto-PGD | 90.4% | **90.6%** | **90.6%** | 90.1% | 84.9% | 88.7% | 82.2% |

Table 7: White box attack success rate for VGG-16 SNN on CIFAR-10 and VGG-11 SNN on CIFAR-100 with HIRE adversarial training method with respect to different surrogate gradients.

| | Arctan | PWL | Erfc | Sigmoid | PWE | Rectangle | Fast Sigmoid | STDB |
|---|---|---|---|---|---|---|---|---|
| | CIFAR-10 | | | | | | | |
| MIM | 83.2% | 66.8% | 67.7% | 42.4% | 16.9% | 47.3% | 79.3% | **94.8%** |
| PGD | 66.7% | 49.6% | 49.7% | 45.3% | 17.6% | 35.6% | 84.9% | **96.4%** |
| AutoPGD | 91.1% | 80.7% | 80.7% | 65.8% | 31.3% | 69.1% | 93.6% | **98.5%** |
| | CIFAR-100 | | | | | | | |
| MIM | 93.5% | **93.8%** | 93.5% | 29.8% | 18.0% | 93.0% | 74.4% | 65.8% |
| PGD | 95.0% | **95.1%** | 94.7% | 30.4% | 19.4% | 94.1% | 78.4% | 69.1% |
| AutoPGD | 95.4% | **95.9%** | 95.8% | 37.9% | 26.2% | 94.9% | 80.8% | 68.0% |

For example, for CIFAR-10, for the DM SNN, the difference in attack success rate between the best and worst SG is 21.9%. If the worst estimator was used, the attack success rate would be 34.1%, whereas if the best estimator was used, the attack success rate is 56.0%. Similarly, for HIRE on CIFAR-100, the attack success rate would be 69.7% higher with the best SG compared to the worst SG. If the worst estimator was used, the attack success rate would be 26.2% compared to 95.9% with the best SG. Just like obfuscating gradients gives false robustness Athalye et al. (2018b), improper surrogate gradients can also yield a false sense of security. Our results clearly show the choice of estimator significantly impacts the success of white-box attacks on

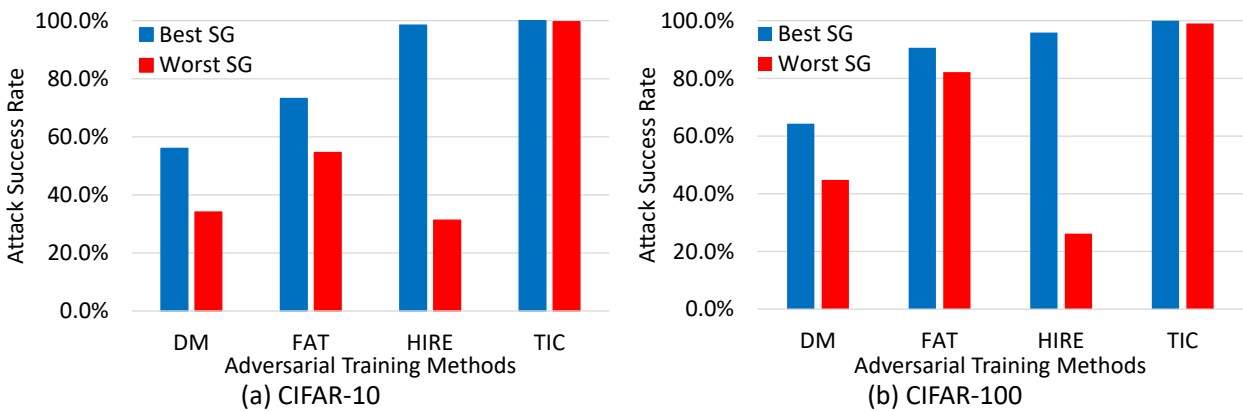

Figure 3: Attack success rate of Auto-PGD with $\epsilon = 0.031$ on adversarially trained SNNs using the best and worst possible Surrogate Gradient Estimator (SG).

Table 8: White box attack success rate for ResNet-19 SNN model with TIC adversarial training method on CIFAR-10, CIFAR-100 with respect to different surrogate gradients.

| | CIFAR-10 | | | | | | |
| --- | --- | --- | --- | --- | --- | --- | --- |
| | Arctan | PWL | Erfc | Sigmoid | PWE | Rectangle | Fast Sigmoid |
| MIM | 99.8% | 99.8% | 99.8% | 99.8% | 99.3% | 99.8% | 97.5% |
| PGD | 100.0% | 99.9% | 100.0% | 100.0% | 99.7% | 99.9% | 98.3% |
| Auto-PGD | 100.0% | 100.0% | 100.0% | 100.0% | 100.0% | 100.0% | 99.7% |
| | CIFAR-100 | | | | | | |
| | Arctan | PWL | Erfc | Sigmoid | PWE | Rectangle | Fast Sigmoid |
| MIM | 99.6% | 99.5% | 99.6% | 99.6% | 98.8% | 99.6% | 94.7% |
| PGD | 99.7% | 99.7% | 99.7% | 99.7% | 99.3% | 99.6% | 96.3% |
| Auto-PGD | 100.0% | 100.0% | 100.0% | 100.0% | 99.8% | 99.9% | 99.1% |

adversarially trained SNNs, highlighting the need for careful selection of SGs to ensure accurate evaluation of model robustness.

## 4 SNN Transferability Study

In this section, we investigate two fundamental security questions pertaining to SNNs:

1. *How vulnerable are SNNs to adversarial examples generated from other machine learning models like Vision Transformers and CNNs?*

2. *Do non-SNN models misclassify adversarial examples created from different types of SNNs?*

Formally, transferability is the phenomenon that occurs when adversarial examples generated using one model are also misclassified by a different model. Transferability studies have been done with CNNs Szegedy et al. (2013); Liu et al. (2016) and with ViTs Mahmood et al. (2021b). To the best of our knowledge, the analysis of the transferability of adversarial examples with respect to SNNs has never been done. Both transfer questions posed at the start of this section, are important from a security perspective. If adversarial samples do not transfer in either direction, then either new SNN/CNN/ViT ensemble defenses are possible. In addition, if adversarial samples between different model types exhibit low transferability, new white-box attacks must be developed to be able to successfully attack both SNNs and non-SNNs simultaneously.

We briefly define how the transferability between different models is measured. Consider a white-box attack $A$ on classifier $C_i$ which produces adversarial example: $x_{adv} = A_{C_i}(x,t)$, where $x$ is the original clean example and $t$ is the corresponding correct class label. Now consider a second classifier $C_j$ independent from classifier $C_i$. The adversarial example $x_{adv}$ transfers from $C_i$ to $C_j$ if and only if the original clean example $x$ is correctly identified by $C_j$ and $x_{adv}$ is misclassified by $C_j$:

$$\{C_j(x) = t\} \land \{C_j(x_{adv}) \neq t\} \tag{19}$$

We can further expand Equation 19 to consider multiple ($n$) adversarial examples:

$$T_{i,j} = \frac{1}{n} \sum_{k=1}^{n} \begin{cases} 1 & \text{if } C_j(A_{C_i}(x_k, t_k)) \neq t_k, \\ 0 & \text{otherwise.} \end{cases} \tag{20}$$

From Equation 20, we can see that a high transferability suggests models share a security vulnerability, that is, most of the adversarial examples generated by $A_{C_i}$ are misclassified by both models $C_i$ and $C_j$.

### 4.1 Transferability Experiment and Analyses

**Experimental Setup:** For our transferability experiment, we analyze four common white-box adversarial attacks which have been experimentally verified to exhibit transferability Mahmood et al. (2021a; 2022). The

Table 9: Transferability results for CIFAR-10. The first column in represents the model used to generate the adversarial examples, $C_i$. The top row in represents the model used to evaluate the adversarial examples, $C_j$. Each entry is the maximum transferability computed using $C_i$ and $C_j$ over four different white-box attacks, Auto-PGD, MIM, PGD and FGSM using Equation 20. Transferability results for other datasets are given in the appendix. Model abbreviations are used for succinctness, S=SNN, R=ResNet, V=VGG-16, C=CNN, BP=Backpropagation, T denotes the Transfer SNN model with corresponding timestep and V=ViT.

| | S-R-BP | S-V-BP | S-V-T5 | S-V-T10 | S-R-T5 | S-R-T10 | VB32 | VB16 | VL16 | C-V | C-R | R101x3 |
|---|---|---|---|---|---|---|---|---|---|---|---|---|
| S-R-BP | 92.00% | 19.30% | 18.30% | 17.10% | 21.10% | 18.00% | 8.70% | 5.60% | 4.80% | 19.60% | 20.10% | 5.00% |
| S-V-BP | 15.30% | 89.90% | 46.20% | 46.60% | 51.80% | 51.50% | 10.10% | 9.80% | 6.50% | 44.00% | 52.30% | 12.20% |
| S-V-T5 | 14.20% | 45.10% | 60.10% | 96.80% | 54.90% | 55.80% | 8.70% | 9.20% | 6.50% | 76.10% | 53.40% | 13.30% |
| S-V-T10 | 13.60% | 42.40% | 98.00% | 57.60% | 52.90% | 52.30% | 8.50% | 9.10% | 6.30% | 73.70% | 51.50% | 12.10% |
| S-R-T5 | 10.10% | 25.50% | 29.70% | 29.50% | 48.70% | 85.30% | 4.10% | 4.40% | 3.70% | 28.60% | 57.50% | 6.60% |
| S-R-T10 | 11.70% | 38.80% | 47.10% | 48.90% | 97.80% | 68.40% | 8.80% | 8.50% | 6.40% | 41.60% | 79.30% | 12.80% |
| VB32 | 10.70% | 14.40% | 15.60% | 15.50% | 21.90% | 20.50% | 100.00% | 83.70% | 75.10% | 13.00% | 20.30% | 60.40% |
| VB16 | 8.90% | 11.90% | 11.70% | 11.50% | 18.90% | 17.40% | 57.40% | 100.00% | 88.90% | 10.60% | 16.80% | 42.90% |
| VL16 | 8.10% | 10.00% | 13.40% | 14.10% | 19.60% | 16.70% | 55.30% | 87.00% | 99.00% | 9.90% | 15.20% | 44.20% |
| C-V | 14.40% | 65.40% | 98.10% | 98.60% | 78.80% | 82.50% | 13.80% | 14.90% | 10.90% | 83.90% | 83.10% | 21.50% |
| C-R | 15.20% | 67.20% | 74.60% | 74.00% | 98.30% | 99.10% | 15.40% | 20.00% | 13.70% | 82.20% | 98.30% | 29.30% |
| R101x3 | 8.50% | 7.30% | 7.10% | 7.50% | 11.80% | 9.80% | 8.60% | 20.00% | 12.30% | 6.10% | 9.70% | 100.00% |

four attacks are the Fast Gradient Sign Method (FGSM) Goodfellow et al. (2015), Projected Gradient Descent (PGD) Madry et al. (2018) the Momentum Iterative Method (MIM) Dong et al. (2018) and Auto-PGD Croce & Hein (2020). For each attack, we use the $l_\infty$ norm with $\epsilon = 0.031$. For brevity, we only list the main attack parameters here and give detailed descriptions of the attacks in the appendix. When running the attacks on SNN models, we use the surrogate gradient function that worked most effectively across all the SNN models for the CIFAR datasets (Arctan) as demonstrated in Section 3. In terms of datasets, we show results for CIFAR-10. When running the transferability experiment between two models, we randomly select 1000 clean examples that are correctly identified by both models and class-wise balanced.

**Models:** To study the transferability of SNNs in relation to other models, we use a wide range of classifiers. These include Vision Transformers: ViT-B-32, ViT-B-16 and ViT-L-16 Dosovitskiy et al. (2020). We also employ a diverse group of CNNs: VGG-16 Simonyan & Zisserman (2014), ResNet-20 He et al. (2016) and BiT-101x3 Kolesnikov et al. (2020). For SNNs, we use both BP and Transfer trained models. For BP SNNs, we experiment with BP SNN VGG-16 Fang et al. (2020) and SEW ResNet Fang et al. (2021a). For Transfer based SNNs we study an SNN VGG-16 Rathi & Roy (2021). All the SNNs are trained following the conventional settings based on their papers. The timesteps we use for different models and corresponding clean accuracies for CIFAR-10 are given in Table 10.

**Experimental Analysis:** The results of our transferability study for CIFAR-10 are visually presented in Figure 4 and corresponding numerical tables are given in the appendix. In Figure 4, each bar corresponds to the maximum transferability attack result measured across Auto-PGD, MIM, FGSM and PGD for the two models. The x-axis of Figure 4 corresponds to the model used to generate the adversarial example ($C_i$ in Equation 20) and the y-axis corresponds to the model used to classify the adversarial example ($C_j$ in Equation 20). Lastly in Figure 4, the colored bars corresponds to the transferability measurements ($T_{i,j}$ in Equation 20). A higher bar means that a large percentage of the adversarial examples are misclassified by both models. Due to the unprecedented scale of our study (12 models with 576 transferability measurements), the results shown in Figure 4 reveal many interesting trends. We summarize the main trends here (and discuss other findings in the appendix):

1. *All types of SNNs and ViTs have remarkably low transferability.* In Figure 4, the yellow bars represent the transferability between BP SNNs and ViTs and the orange bars represent the transferability between Transfer SNNs and ViTs. We can clearly see adversarial examples do not transfer between the two. For example, the SEW ResNet (S-R-BP) misclassifies adversarial examples generated by ViT-L-16 (V-L16) only 8.1% of the time. Likewise, across all ViT models that evaluate adversarial examples created by SNNs, the transferability is also low. The maximum transferability for this type of pairing occurs between ViT-B-32 (V-B32) and the Backprop SNN VGG (S-V-BP) at a low 10.1%.

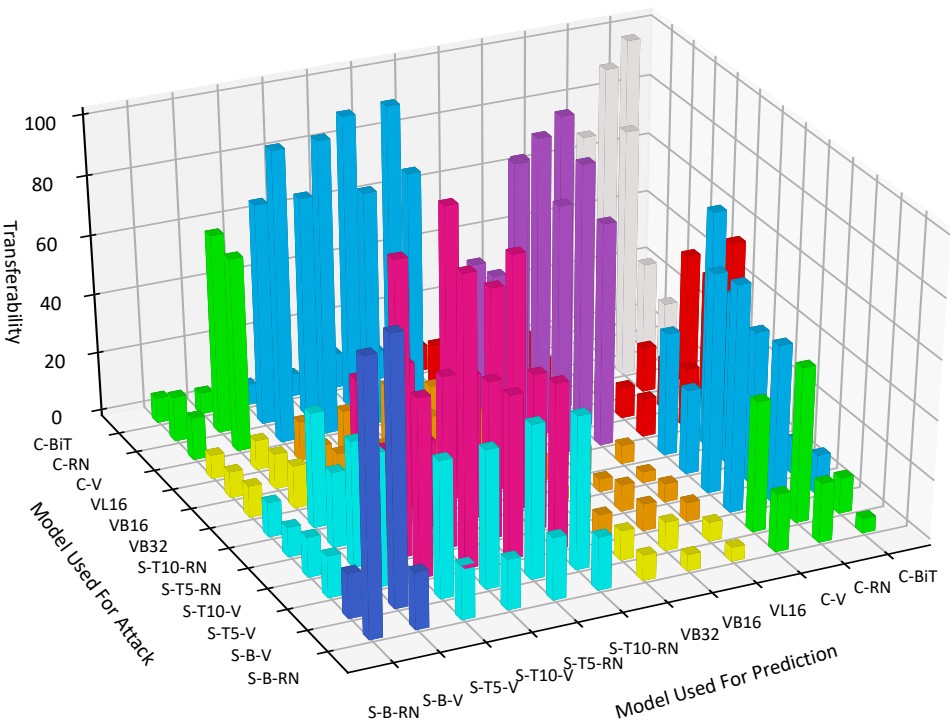

Figure 4: Visual representation of transferability results for CIFAR-10. Model abbreviations are used for succinctness, S=SNN, R=ResNet, V=VGG-16, C=CNN, BP=Backpropagation, T denotes the Transfer SNN model with corresponding timestep and V-L=ViT-L.

2. *Transfer SNNs and CNNs have high transferability, but BP SNNs and CNNs do not.* In Figure 4, the blue bars represent the transferability between Transfer SNNs and CNNs, which we can visually see is large. For example, 99.1% of the time the Transfer SNN ResNet with timestep 10 (S-R-T10) misclassifies adversarial examples created by the CNN ResNet (C-R). This is significant because it highlights that when weight transfer training is done, both SNN and CNN models still share the same vulnerabilities. The exception to this trend is the CNN BiT-101x3 (C-101x3). We hypothesize that the low transferability of this model with SNNs occurs due to the difference in training (BiT-101x3 is pre-trained on ImageNet-21K and uses a non-standard image size (160x128) in our implementation).

Overall, our transferability study shows that there exists multiple model pairings between SNNs, ViTs and CNNs that exhibit the low transferability phenomena for Auto-PGD, MIM and PGD and FGSM. *This is a critical finding because it demonstrates that single model white-box attacks are not effective across SNN and non-SNN models simultaneously.*

## 5 Mixed Dynamic Spiking Estimation Attack

We have demonstrated two major issues currently facing white-box adversarial attacks on SNNs. First, in Section 3, we showed the choice of surrogate gradient estimator heavily influences how successful the attack is. While certain estimator functions performed better than others, the result was highly dependent on the dataset and model. There was no single estimator that worked best across all models and all datasets. The second major issue is that even when an effective gradient estimator is employed, adversarial examples created by attacks like Auto-PGD are not misclassified by SNNs and non-SNN models simultaneously. We demonstrated this result in Section 4. State-of-the-art white-box attacks do not transfer well and do not exploit the capabilities of different gradient estimators for SNNs. To address the two major problems, we propose a new white-box attack, the Mixed Dynamic Spiking Estimation (MDSE) Attack. Our new attack is

comprised of two main components, dynamic estimation of the spiking gradients and mixing of the gradients when attacking multiple models. The pseudo-code for the MDSE attack is given in Algorithm 1.

## 5.1 Dynamic Gradient Estimation

In general, an SNN white-box adversarial example is iterative created from clean example $(x, y)$: $x_{adv}^{(i+1)} = x_{adv}^{(i)} + A(u_k^{(i)}(\cdot), \frac{\partial L(u_k^{(i)}(\cdot))}{\partial x_{adv}^{(i)}}, \epsilon_s)$, where $A$ denotes the attack algorithm (e.g. Auto-PGD), $u_k^{(i)}(\cdot)$ is the $k^{th}$ gradient estimator from the set of all possible SNN surrogate gradient estimators $U$ and $\epsilon_s$ is the amount of perturbation added in the $i^{th}$ step of the attack. For SNNs, we first demonstrated that the choice of gradient estimator has a huge impact on attack success rate in Section 3. However, all these white-box attacks were done using homogeneous estimators, i.e., for an $N$ step attack: $\forall_{i=1}^{N-1} u^{(i)}(\cdot) = u^{(i+1)}(\cdot)$. For SNN models, multiple surrogate gradient estimators exist, meaning an attack can utilize any $u \in U$ at each iteration of the attack in a heterogeneous manner. In our new attack formulation, we exploit the fact that different surrogate gradient estimators can give higher fidelity results for different models and samples, by employing a dynamic gradient estimation scheme. Specifically, for each attack step $A$, the $i^{th}$ gradient is computed using surrogate gradient function $u_k^{(i)}$ through maximization of the loss function:

$$u_k^{(i)} = \max_{u \in U} L(x_{adv}^{(i)} + A(u(\cdot), \frac{\partial L(u(\cdot))}{\partial x_{adv}^{(i)}}, \epsilon_s)) \tag{21}$$

## 5.2 Mixed Multi-Model Attack

A critical issue for current white-box attacks is that they do not create adversarial examples that are transferable between SNNs and non-SNNs. Hence, a single white-box attack is not effective against an ensemble of SNN and non-SNN models. To rectify this problem, we propose a framework that leverages the gradients from multiple models, the "mixed" part of the Mixed Dynamic Spiking Estimation (MDSE) attack. In MDSE the adversarial sample is computed iteratively:

$$x_{adv}^{(i+1)} = x_{adv}^{(i)} + \epsilon_{step} * \text{sign}(G_{blend}(x_{adv}^{(i)})) \tag{22}$$

where $x_{adv}^{(1)} = x$ and $\epsilon_{step}$ is the step size for each iteration of the attack. The difference between a single model attack like Auto-PGD and MDSE lies in the value of $G_{blend}$:

$$G_{blend}(x_{adv}^{(i)}) = \sum_{s \in S} \alpha_s \phi_s(x_{adv}^{(i)}, U) + \sum_{v \in V} \alpha_v \beta_v \odot \frac{\partial L_v}{\partial x_{adv}^{(i)}} \tag{23}$$

where $S$ represents the set of all SNN models being attacked, $\alpha_s$ represent the weighting coefficient of the gradient associated with SNN model $s$ and $V$ represents the set of CNN and ViT models being attacked. In Equation 23, $\phi_s(\cdot)$ represents the dynamic gradient estimation (from Equation 21) with respect to model $s$ such that $\phi_s(x_{adv}^{(i)}, U) = \frac{\partial L_s(u_k(\cdot))}{\partial x_{adv}^{(i)}}$. Lastly in Equation 23, $\beta_v$ denotes the attention-roll out term which is computed based on the attention weight matrix for Transformer models Mahmood et al. (2021b) and is simply a matrix of ones ($J$), for non-ViT models. For the $m^{th}$ model, the weight coefficient $\alpha_m$ from Equation 23 is automatically computed in each iteration of the attack:

$$\alpha_m^{(i+1)} = \alpha_m^{(i)} - r \frac{\partial F}{\partial \alpha_m^{(i)}} \tag{24}$$

where $r$ is the learning rate for the coefficients and the effectiveness of the coefficients is measured and updated based on a modified version of the non-targeted loss function proposed in Carlini & Wagner (2017):

$$F(x_{adv}^{(i)}) = \max(z(x_{adv}^{(i)})_t - \max\{(z(x_{adv}^{(i)})_j : j \neq t\}, -\kappa) \tag{25}$$

where $z(\cdot)_j$ represents the $j^{th}$ softmax output (probability) from the model, $z(\cdot)_t$ represents the softmax probability of the correct class label $t$ and $\kappa$ represents confidence with which the adversarial example should be

---

**Algorithm 1 Mixed Dynamic Spiking Estimation Attack**

---

1: **Input**: clean sample $x$, number of iterations $N$, step size per iteration $\epsilon_{step}$, maximum perturbation $\epsilon_{max}$, set of SNN models $S$, set of non-SNN models $V$, corresponding loss function $L_1, .., L_M$ for all $M$ models and coefficient learning rate $r$.

2: $x_{adv}^{(0)} = x$

3: **For** $i$ in range 1 to $N$ do:

4:      *//Generate the adversarial example*

5:      $G_{blend}(x_{adv}^{(i)}) = \sum_{s \in S} \alpha_s \phi_s(x_{adv}^{(i)}, U) + \sum_{v \in V} \alpha_v \beta_v \odot \frac{\partial L_v}{\partial x_{adv}^{(i)}}$

6:      $x_{adv}^{(i+1)} = x_{adv}^{(i)} + \epsilon_{step} \text{sign}(G_{blend}(x_{adv}^{(i)}))$

7:      *//Apply projection operation*

8:      $x_{adv}^{(i+1)} = P(x_{adv}^{(i+1)}, x, \epsilon_{max})$

9:      **For** $i$ in range 1 to $M$ do:

10:         *//Update the model coefficients, note $\beta_m = J$ for every non-ViT model*

11:         $\frac{\partial x_{adv}^{(i)}}{\partial \alpha_m^{(i)}} \approx \sigma \epsilon_{step} \text{sech}^2(\sigma \sum_{m=1}^{M} \frac{\partial L_m}{\partial x_{adv}^{(i)}} \odot \beta_m) \odot \frac{\partial L_m}{\partial x_{adv}^{(i)}} \odot \beta_m$

12:         $\frac{\partial F}{\partial \alpha_m^{(i)}} = \frac{\partial F}{\partial x_{adv}^{(i)}} \odot \frac{\partial x_{adv}^{(i)}}{\partial \alpha_m^{(i)}}$

13:         $\alpha_m^{(i+1)} = \alpha_m^{(i)} - r \frac{\partial F}{\partial \alpha_m^{(i)}}$

14:      end for

15: end for

16: **Output**: $x_{adv}$

---

misclassified (in our attacks, we use $\kappa = 0$). Equation 24 can be computed by expanding $\frac{\partial F}{\partial \alpha_m^{(i)}} = \frac{\partial F}{\partial \alpha_{adv}^{(i)}} \odot \frac{\partial x_{adv}^{(i)}}{\partial \alpha_m^{(i)}}$ and approximating the derivative of $\text{sign}(x)$ in Equation 22 with $\sigma \cdot \text{sech}^2(\sigma x)$:

$$\frac{\partial x_{adv}^{(i)}}{\partial \alpha_m^{(i)}} \approx \sigma \epsilon_{step} \text{sech}^2(\sigma \sum_{m=1}^{M} \frac{\partial L_m}{\partial x_{adv}^{(i)}}) \odot \frac{\partial L_m}{\partial x_{adv}^{(i)}} \tag{26}$$

where $u$ is a fitting factor for the derivative approximation.

**Advantages of MDSE:** There are several pertinent advantages of the MDSE attack over other white-box attacks. The Self-Attention Gradient Attack (SAGA) was proposed in Mahmood et al. (2021b) for attacking multiple models, similar to MDSE. However, in regards to non-SNN models, SAGA has two key limitations that MDSE overcomes. Assume a model ensemble containing the set of models $E = S \cup V$ and $|E| = M$. Every model $m$ requires its own weighting factor such that $\overrightarrow{\alpha} = (\alpha_1, \cdots, \alpha_m, ..., \alpha_M)$. If these hyperparameters are not properly chosen, the attack performance of SAGA degrades significantly. This was demonstrated in Mahmood et al. (2021b). In MDSE, these coefficients are adaptively updated at every iteration of the attack, removing this pitfall. The second drawback of SAGA is that once $\overrightarrow{\alpha}$ is chosen for the attack, it is fixed for every sample and for every iteration of the attack. This makes choosing $\overrightarrow{\alpha}$ incredibly challenging as each scalar hyperparameter $\alpha_m$ must either perform well for the majority of samples or have to be manually selected on a per sample basis (since $\alpha_m \in \mathbb{R}$). In MDSE, $\overrightarrow{\alpha}$ is fine grained on a per sample and per pixel basis i.e., $\alpha_m \in \mathbb{R}^{b \times h \times w \times c}$ where $b \times h \times w \times c$ represent the batch size, height, width and number of color channels for the input to MDSE. In addition to overcoming two key limitations of SAGA for non-SNN models, MDSE also leverages the dynamic gradient estimation scheme. This makes MDSE stronger for ensembles that include SNNs, something SAGA lacks.

## 6 Experimental Results

### 6.1 Experimental Setup

All experiments are conducted in PyTorch using a workstation equipped with an AMD Ryzen Threadripper PRO 3975WX 32-core processor, 256 GB of memory, and two NVIDIA GeForce RTX 3090Ti GPUs. To

Table 10: Clean Accuracy and timesteps for models for CIFAR-10, CIFAR-100, ImageNet datasets.

| | Model | Timesteps | Accuracy | | Model | Timesteps | Accuracy |
|---|---|---|---|---|---|---|---|
| | S-R-BP | 4 | 81.10% | | S-R-BP | 5 | 65.10% |
| | S-V-BP | 20 | 89.20% | | S-V-BP | 30 | 64.10% |
| | S-V-T5 | 5 | 90.90% | | S-V-T10 | 10 | 65.40% |
| | S-V-T10 | 10 | 91.40% | CIFAR-100 | S-R-T8 | 8 | 59.70% |
| | S-R-T5 | 5 | 89.20% | | C-101x3 | - | 91.80% |
| | S-R-T10 | 10 | 91.60% | | C-V | - | 66.60% |
| CIFAR-10 | C-101x3 | - | 98.70% | | V-L16 | - | 94.00% |
| | C-V | - | 91.90% | | S-R-BP | 4 | 60.82% |
| | C-R | - | 92.10% | | S-V-T5 | 5 | 57.53% |
| | V-L16 | - | 99.10% | ImageNet | C152x4-512 | - | 85.31% |
| | V-B32 | - | 98.60% | | C-V | - | 71.59% |
| | V-B16 | - | 98.90% | | V-L16 | - | 82.94% |

evaluate the attack performance of MDSE, we conducted experiments on CIFAR-10, CIFAR-100 and ImageNet datasets. We test 13 different pairs of models for CIFAR-10/CIFAR-100 and 7 pairs of models for ImageNet. For the ImageNet models, we include the Vision Transformer (V-L-16), Big Transfer CNN (C152x4-512) with corresponding input image size $512 \times 512$ and VGG-16 (C-V). We also use a BP trained SNN ResNet-18 (S-R-BP) and a VGG-16 Transfer-based SNN (S-V-T5). Clean accuracy for each model and detailed timesteps for each SNN model are provided in Table 10.

In addition to attacking undefended model pairs with low transferability, we also evaluate MDSE against various pairs of adversarially trained SNNs for CIFAR-10 and CIFAR-100. Similar to the gradient estimator experiments, we employ four adversarial training methods (FAT, DM, HIRE, and TIC) with their corresponding clean accuracy and timesteps provided in Table 11. We further include two SOTA adversarial trained SNNs from Liu et al. represented as SR* for sparsity regularization strategy with adversarial training. We use the given checkpoints on VGG-11 (SR*-V11), and WideResNet Zagoruyko (2016) with a depth of 16 and width of 4 (SR*-W16) for CIFAR-10 in evaluation.

To attack each model pair, we use 1000 correctly identified class-wise balanced samples from the validation set. For the attack, we use a maximum perturbation of $\epsilon = 0.031$ for CIFAR datasets and $\epsilon = 0.062$ for ImageNet with respect to the $l_\infty$ norm. We compare MDSE to the Auto-PGD, MIM, PGD and SAGA attacks. We generally use batch size 50 for all the attacks and reduce it if the GPU memory is insufficient.

1. For single MIM, PGD, and Auto-PGD attacks, we use attack steps = 40 to generate AEs from each model. For MIM and PGD, we set attack step $\epsilon_{step} = 0.005$ or 0.01. We use Auto-PGD on the cross-entropy. For the single model's attacks (e.g. Auto-PGD), we use the the highest attack success rate on each pair of models, which we denote as "Max Auto".

2. For SAGA, we set the attack as a balanced version of SAGA that uses coefficients $\alpha_1 = \alpha_2 = 0.5$ for two models to generate AEs and get the attack success rate among both models.

3. For MDSE, we set the learning rate $r = 10,000$ or $100,000$ for the coefficients. We set attack steps = 40 and $\epsilon_{step} = 0.005$ or 0.01 to generate AEs and get the attack success rate among both models.

Table 11: Clean Accuracy and timesteps for adversarial trained SNNs for CIFAR-10, CIFAR-100 datasets.

| | Model | TIC-R19 | HIRE-V16 | DM-R18 | FAT-R18 | SR*-V11 | SR*-W16 |
|---|---|---|---|---|---|---|---|
| CIFAR10 | Timesteps | 10 | 8 | 5 | 5 | 8 | 8 |
| | Accuracy | 92.3% | 89.0% | 66.8% | 73.2% | 85.9% | 85.6% |
| | Model | TIC-R19 | HIRE-V11 | DM-R18 | FAT-R18 | | |
| CIFAR100 | Timesteps | 10 | 8 | 5 | 5 | | |
| | Accuracy | 72.1% | 66.1% | 41.0% | 40.8% | | |

For these attacks, we utilize the optimal SG studied in the preceding sections. Our MDSE approach incorporates Arctan, PWL, and Erfc for all SNNs, and additionally integrates various SGs, such as Sigmoid, PWE, Rectangle, Fast Sigmoid, and STDB, tailored to different SNNs to demonstrate their attack capabilities effectively. In these experiments, the attack success rate is the percent of adversarial examples that are misclassified by *both* models in the pair of models. We run each attack with a combination of SGs settings and present the best results among them.

**Ensemble Attack Success Rate** - In the context of proposing a new attack, it is important to define what constitutes a "successful" attack so that attack success rate can be measured, and different white-box attacks can be directly compared. It is important to note that in the literature there are two established methodologies for measuring the attack success rate on model ensembles. We denote each of these methodologies as *any* and *all*. We will first mathematically define these and then justify our choice of attack measurement.

When attacking an ensemble of models we can consider this group as $\Delta = S \cup V$ where $S$ is the set of SNN models and $V$ is the set of non-SNN models. An adversarial example $x_{adv}$ with corresponding clean class label $y$ is considered a successful adversarial attack under the *any* attack metric if the following condition holds: $\exists c \in \Delta, \text{s.t.} c(x_{adv}) \neq y$. Essentially this means under the *any* metric, as long as the adversarial example is misclassified by any of the models in the ensemble the attack is considered successful. This metric has been used in previous literature including Ozdenizci & Legenstein; Liu et al.. The *all* metric defines a successful adversarial attack as follows: $\forall c \in \Delta, c(x_{adv}) \neq y$. Under this metric an adversarial example is only considered a successful attack if all models in the ensemble produce the wrong class label. This metric has been adopted in many works including Liu et al. (2016); Mahmood et al. (2021b).

The *all* metric has the following three advantages. First, this metric accurately reflects attack success rate even when majority voting is used in ensembles. Under the *all* condition, even if the defender attempts to form a consensus from model voting, no class label is correct so the defender will never be able to predict the correct class label from the ensemble outputs. Second, the *all* metric reflects the worst case scenario for the attacker, which presents a more realistic lower bound on the attack performance. If even one model in the ensemble has $c(x_{adv}) = y$ then the assumption of this metric is that the defender picks that model and deduces the correct class. It is a standard practice in security to assume the worst case for the defender (when proposing a new defense) Carlini et al. (2019) and the worst case for the attacker (when proposing a new attack). Assuming a stronger defender gives a lower bound on the performance of the attack. Third and lastly, the *any* metric suffers from an issue which we denote as the "weakest model link" that the *all* metric does not suffer from. Assume a set model ensemble $S$ contains $n$ models where $n \geq 2$. Let us denote a weak model $c_w$ where $c_w \in \Delta$ and $\forall x_{adv} \in X, c_w(x_{adv}) \neq y$ where $X$ is the set of all adversarial examples on which we wish to measure the attack success rate. Further consider the case where one or more models in set $S$ are not successfully attacked, $\forall x_{adv} \in X, \exists c \in \Delta, \text{s.t.} \{c(x_{adv}) = y\}$. Under the *any* metric, the attack success rate would be reported as 100%, even though at least one model in the ensemble correctly identified $x_{adv}$. In reality, the defender still has probability $p \geq \frac{1}{n}$ of picking the right class label if randomly selecting between ensemble classifiers, when classifiers do not all return the same class label. In short, by adding a weak model $c_w$ to any ensemble, the attack success rate can be artificially boosted if the *any* metric is used. For all of these reasons we use the *all* metric when measuring attack success rate for each attack in our analyses in the experiments.

## 6.2 Experimental Analyses

Figure 5 compares the attack success rates of MIM, PGD, Auto-PGD, SAGA and MDSE attacks for different model pairs across CIFAR-10, CIFAR-100, and ImageNet. Each figure is sorted in decreasing order based on MDSE results. The results indicate that MDSE consistently achieves the highest attack success rate across all datasets and model pairs. Other attacks are not effective and perform inconsistently on different model pairs. Notably, MDSE significantly outperforms other attacks with high accuracy while single-model attacks and SAGA show limited effectiveness, especially in small datasets like CIFAR-10.

Table 12: Max MIM, PGD, and Auto represent the max success rate using adversarial examples generated by model 1 and model 2 for CIFAR-10, CIFAR-100.

| Model 1 | Model 2 | Max MIM | Max PGD | Max Auto | SAGA | **MDSE** |
|---------|---------|---------|---------|----------|------|----------|
| C-V | S-R-BP | 18.5% | 16.1% | 15.8% | 26.6% | **90.4%** |
| C-V | S-V-BP | 72.7% | 74.3% | 75.8% | 81.4% | **99.5%** |
| C-V | S-V-T10 | 88.6% | 89.2% | 90.7% | 87.6% | **90.7%** |
| C-V | S-R-T10 | 86.6% | 87.3% | 88.8% | 77.3% | **91.4%** |
| S-R-BP | S-V-T10 | 15.3% | 13.4% | 12.4% | 18.4% | **73.4%** |
| V-L16 | S-R-BP | 12.5% | 10.7% | 8.9% | 23.9% | **96.8%** |
| V-L16 | S-V-BP | 10.7% | 7.1% | 6.4% | 52.4% | **97.8%** |
| V-L16 | S-V-T10 | 9.5% | 4.8% | 4.8% | 28.4% | **92.7%** |
| V-L16 | S-R-T10 | 16.0% | 7.7% | 8.6% | 36.6% | **99.0%** |
| C-101x3 | S-R-BP | 17.3% | 14.3% | 12.3% | 58.7% | **95.7%** |
| C-101x3 | S-V-BP | 15.3% | 8.9% | 8.5% | 31.6% | **95.3%** |
| C-101x3 | S-V-T10 | 22.2% | 15.2% | 7.1% | 30.2% | **98.0%** |
| C-101x3 | S-R-T10 | 25.4% | 16.8% | 7.7% | 62.3% | **98.8%** |

(a) CIFAR-10

| Model 1 | Model 2 | Max MIM | Max PGD | Max Auto | SAGA | **MDSE** |
|---------|---------|---------|---------|----------|------|----------|
| C-V | S-R-BP | 40.7% | 33.6% | 40.4% | 49.8% | **93.2%** |
| C-V | S-V-BP | 59.4% | 51.3% | 57.6% | 67.2% | **94.7%** |
| C-V | S-V-T10 | 73.1% | 68.6% | 70.3% | 78.6% | **84.0%** |
| C-V | S-R-T8 | 69.6% | 46.6% | 68.5% | 84.4% | **91.8%** |
| S-R-BP | S-V-T10 | 41.7% | 33.7% | 29.8% | 45.3% | **64.3%** |
| V-L16 | S-R-BP | 28.3% | 23.5% | 22.1% | 74.5% | **78.9%** |
| V-L16 | S-V-BP | 33.9% | 20.3% | 18.8% | 70.0% | **85.4%** |
| V-L16 | S-V-T10 | 25.7% | 15.3% | 13.6% | 33.0% | **91.5%** |
| V-L16 | S-R-T8 | 27.2% | 17.4% | 15.3% | 60.8% | **93.8%** |
| C-101x3 | S-R-BP | 38.3% | 32.6% | 30.3% | 52.0% | **77.3%** |
| C-101x3 | S-V-BP | 22.7% | 16.9% | 16.1% | 57.0% | **83.8%** |
| C-101x3 | S-V-T10 | 24.6% | 20.3% | 17.9% | 44.5% | **84.5%** |
| C-101x3 | S-R-T8 | 25.2% | 21.0% | 19.5% | 85.8% | **97.0%** |

(b) CIFAR-100

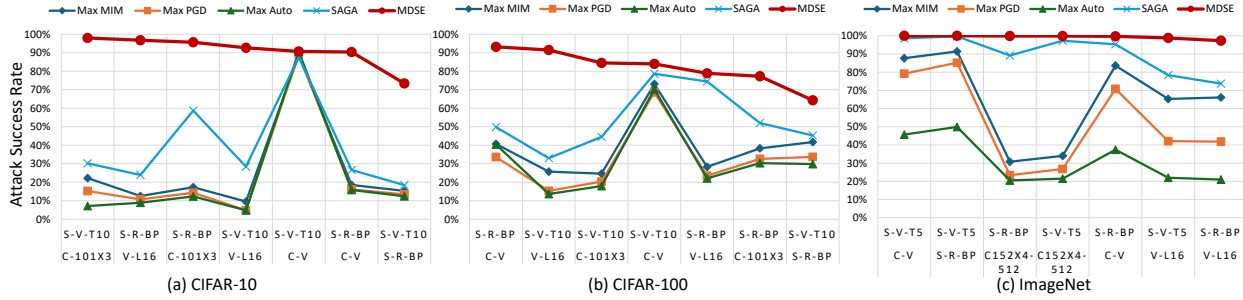

Figure 5: Attack success rates of Max MIM, PGD, Auto-PGD, SAGA and MDSE on CIFAR-10, CIFAR-100 and ImageNet for different pairs of SNN and non-SNN models. Sorted by MDSE results in decreasing order.

### 6.2.1 CIFAR-10

In Table 12(a), we attack 13 different pairs of models, which include different combinations of SNNs, CNNs and ViTs for CIFAR-10. For the pairings of models, there are several novel findings. For pairs that contain an SNN and ViT, MDSE performs well even when all other attacks do not. For example, for CIFAR-10 with

Table 13: Max MIM, PGD, and Auto represent the max success rate using adversarial examples generated by model 1 and model 2 for ImageNet.

| Model 1 | Model 2 | Max MIM | Max PGD | Max Auto | SAGA | **MDSE** |
|---------|---------|---------|---------|----------|------|----------|
| C-V | S-R-BP | 83.6% | 70.8% | 37.4% | 95.3% | **99.7%** |
| C-V | S-V-T5 | 87.7% | 79.2% | 45.7% | 98.6% | **100.0%** |
| S-R-BP | S-V-T5 | 91.4% | 85.2% | 49.9% | 99.7% | **100.0%** |
| V-L16 | S-R-BP | 66.1% | 41.8% | 21.0% | 73.7% | **97.3%** |
| V-L16 | S-V-T5 | 65.3% | 42.1% | 22.0% | 78.4% | **98.8%** |
| C152x4-512 | S-R-BP | 30.8% | 23.4% | 20.5% | 89.2% | **99.9%** |
| C152x4-512 | S-V-T5 | 34.0% | 26.8% | 21.5% | 97.3% | **99.9%** |

ViT-L-16 (V-L16) and the SEW ResNet (S-R-BP), the best non-SAGA result achieves an attack success rate of only 12.5%, whereas MDSE achieves 96.8%. For pairs that contain a CNN and the corresponding Transfer SNN (which uses the CNN weights as a starting point), even single-model attacks like MIM and PGD work well. For example, consider the pair: Transfer SNN VGG-16 (S-V-T10) and CNN VGG-16 (C-V). For CIFAR-10, MIM gives an attack success rate of 88.6% (MDSE achieves 90.7%). This shared vulnerability likely arises from the shared model weights. Lastly, SAGA in general, generates adversarial examples more effectively than the Auto-PGD or MIM attacks. However, its performance is still much worse than MDSE. For example, MDSE has an average attack success rate improvement of 46.6% over SAGA for the CIFAR-10 pairs we tested.

### 6.2.2 CIFAR-100

Table 12(b) shows the attack success rates on 13 different pairs of models for CIFAR-100. Similar to CIFAR-10, MDSE consistently achieves the highest success rates across all tested pairs. However, the attack success rates of single-model attacks and SAGA are relatively higher compared to CIFAR-10 results. This is mainly because as the task becomes more complicated, the clean accuracy is lower, making it easier for adversarial attacks to succeed and have higher transferability. The transferability between SNNs and ViTs remains low for most attacks except for MDSE; for instance, the best attack success rate is only 33.0% with SAGA, while MDSE achieves 91.5%.

### 6.2.3 ImageNet

In Table 13, we attack 7 different pairs of ImageNet models and report the attack success rate. Overall, MDSE's performance for ImageNet shows a similar trend to the CIFAR datasets that work for all pairs with very high attack success rates. In particular, even for the smallest case of attack success rate gap, Transfer SNN ResNet-18 (S-R-BP) and ViT-L-16 (V-L16), MDSE performs 24.8% better than any other white-box attack. Additionally, the results indicate that other attacks, besides MDSE, do not have consistent attack capabilities and may only be effective against specific model pairs. For example, Auto-PGD attacks perform poorly on ImageNet, while MIM attacks show a 61.8% variability in attack success rates between different model pairs.

Overall, the results presented here demonstrate a clear trend. Traditional white-box attacks have a low attack success rate against most pairs that include an SNN and non-SNN model. Therefore, it is imperative to use strong multi-model attacks like MDSE to consistently and effectively evaluate the robustness of SNNs and other models

### 6.2.4 Experiments on Adversarial Trained SNNs

We summarize the attack success rates in Table 14a for CIFAR-10 and Table 14b for CIFAR-100, with the results visualized in Figure 6. Consistent with the trend observed in undefended models, MDSE achieves the highest attack success rate among all pairs of SNNs on both CIFAR-10 and CIFAR-100 datasets.

Table 14: Max MIM, PGD, and Auto represent the max success rate using adversarial examples generated by adversarial trained SNN model 1 and model 2 for CIFAR-10 and CIFAR-100 dataset.

| Model1 | Model2 | Max MIM | Max PGD | Max Auto | SAGA | **MDSE** |
|---|---|---|---|---|---|---|
| TIC-R19 | HIRE-V16 | 56.0% | 57.8% | 48.6% | 38.6% | **68.5%** |
| FAT-R18 | HIRE-V16 | 11.6% | 10.0% | 10.2% | 12.0% | **47.1%** |
| DM-R18 | HIRE-V16 | 7.7% | 7.2% | 8.5% | 13.8% | **38.5%** |
| DM-R18 | FAT-R18 | 18.1% | 16.9% | 22.2% | 21.2% | **29.7%** |
| DM-R18 | SR*-V11 | 16.0% | 15.5% | 18.0% | 18.9% | **27.6%** |
| FAT-R18 | TIC-R19 | 10.2% | 10.3% | 8.9% | 8.4% | **27.1%** |
| DM-R18 | SR*-W16 | 15.3% | 15.1% | 14.7% | 19.5% | **25.8%** |
| DM-R18 | TIC-R19 | 8.6% | 8.6% | 8.1% | 7.1% | **25.4%** |

(a) CIFAR-10

| Model1 | Model2 | Max MIM | Max PGD | Max Auto | SAGA | MDSE |
|---|---|---|---|---|---|---|
| TIC-R19 | HIRE-V11 | 68.5% | 69.0% | 66.1% | 41.0% | **79.3%** |
| FAT-R18 | HIRE-V11 | 28.3% | 29.6% | 28.5% | 21.7% | **54.5%** |
| FAT-R18 | TIC-R19 | 25.2% | 25.5% | 27.1% | 24.2% | **47.7%** |
| DM-R18 | FAT-R18 | 27.9% | 27.7% | 29.5% | 31.3% | **41.0%** |
| DM-R18 | HIRE-V11 | 14.8% | 16.0% | 17.7% | 14.6% | **39.0%** |
| DM-R18 | TIC-R19 | 12.1% | 11.2% | 11.5% | 12.9% | **38.5%** |

(b) CIFAR-100

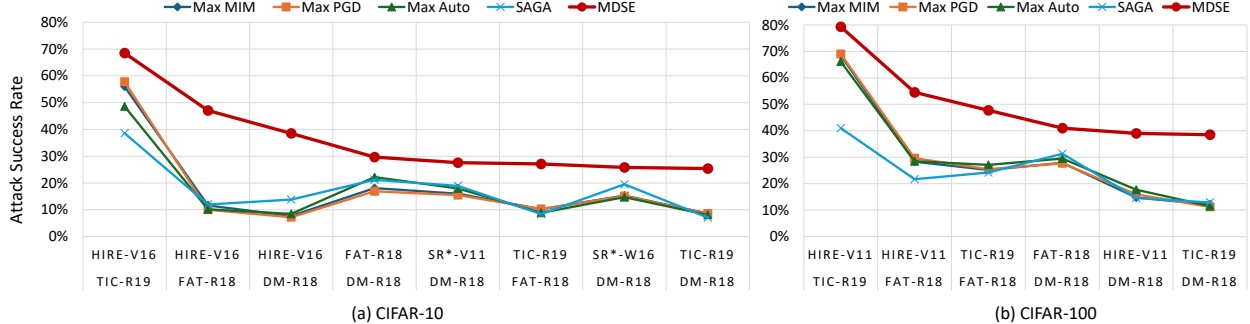

Figure 6: Attack success rates on Max MIM, PGD, Auto-PGD, SAGA and MDSE on CIFAR-10, CIFAR-100 with different adversarial trained SNN pairs. Sorted by MDSE results in decreasing order.

As indicated in Figure 3, some adversarially trained SNNs exhibit enhanced robustness against single-model attacks. We extend this investigation to evaluate the robustness against pairs of adversarially trained SNNs. Interestingly, our findings reveal that while white-box attacks may succeed for single SNNs, combining two adversarially trained SNNs can significantly enhance robustness against these attacks. For example, on CIFAR-10, Auto-PGD achieves attack success rates of 56.0% and 100.0% against DM and TIC trained SNNs, respectively. However, combining these two SNNs reduces the attack success rate to 8.1%. In contrast, MDSE achieves a 25.4% success rate against the same pairing, demonstrating its superior attack effectiveness. Figure 6 shows that, apart from the pair of HIRE-V16 and TIC-R19 which exhibit high attack success rates for most attacks, no other attack achieves a success rate of 20% for CIFAR-10 or 30% for CIFAR-100, except for MDSE.

Overall, our results demonstrate that MDSE is the most effective multi-model attack, even against a two-model adversarially trained defense. This underscores the importance of employing robust multi-model attacks like MDSE to comprehensively evaluate the resilience of adversarially trained SNNs.

Table 15: Max Auto, SAGA, MDSE attack success rate using adversarial examples generated by DM-R18 and FAT-R18 SNN for CIFAR-10 using different number of SGs.

| Number of SGs | AutoPGD | SAGA | MDSE |
|---|---|---|---|
| 1 (Arctan) | 21.7% | 21.0% | 23.2% |
| 4 | 20.1% | 21.0% | 28.4% |
| 5 | 21.4% | 21.2% | 27.6% |
| 7 | 22.2% | 20.4% | 29.7% |

Table 16: Attack success rates on adversarial trained SNNs with proposed attacks using 1 SG (MDS) and multiple SGs (MDSE) for CIFAR-10 and CIAFR-100 .

| | | CIFAR10 | | CIFAR100 | |
|---|---|---|---|---|---|
| Model 1 | Model 2 | MDS with 1 SG | MDSE | MDS with 1 SG | MDSE |
| TIC-R19 | HIRE-V16/V11 | 66.4% | **68.5%** | 72.9% | **79.3%** |
| FAT-R18 | HIRE-V16/V11 | 22.6% | **47.1%** | 50.1% | **54.5%** |
| DM-R18 | HIRE-V16/V11 | 21.4% | **38.5%** | 38.0% | **39.0%** |
| DM-R18 | FAT-R18 | 23.2% | **29.7%** | 37.4% | **41.0%** |
| FAT-R18 | TIC-R19 | 23.8% | **27.1%** | 46.6% | **47.7%** |
| DM-R18 | TIC-R19 | 22.2% | **25.4%** | 35.8% | **38.5%** |

### 6.2.5 Ablation study on impacts of dynamic gradient estimation on MDSE

We further conduct experiments on a select pair of SNNs to examine and demonstrate the effectiveness of the dynamic gradient estimation in the attack. Table 15 shows the attack success rates for AutoPGD, SAGA and MDSE for adversarial examples generated by DM-R18 and FAT-R18 SNN for CIFAR-10 using different numbers of SGs used during the attack. We use a single SG (Arctan), which is generally the best performance surrogate gradient estimator as studied in Section 3. Then we extend the number of SGs to 4 (Arctan, PWL, Erfc, and Rectangle), 5 (Arctan, PWL, Erfc, Rectangle, and Sigmoid), and 7 (Arctan, PWL, Erfc, Rectangle, Sigmoid, PWE, and Fast Sigmoid) to run the attacks. The results indicate that while adding more SGs offers limited improvements for AutoPGD and SAGA attacks, it significantly enhances the attack effectiveness when using MDSE, with more SGs contributing to stronger attacks.

The results in Table 16 display the attack success rates for different adversarially trained SNN pairs using our proposed MDSE attack, with both a single SG and multiple SGs. We can observe that even with the fixed choice of SG, our mixed multi-model attack – featuring adaptively updated coefficients for each model – outperforms other attacks, as seen in comparison with Table 14. Moreover, the attack success rate consistently improves for all model pairs when dynamic gradient estimation with multiple SG options is employed at each attack step. Specifically, we achieve an improvement of approximately 2.1% to 24.5% on CIFAR-10 and 1.0% to 6.4% on CIFAR-100, even against these robust adversarially trained SNNs.

## 7 Conclusion

Developments in SNNs create new opportunities for energy efficiency but also raise critical security questions. In this paper, we investigated three important aspects of SNN adversarial machine learning among BP SNNs, Transfer SNNs, and adversarial trained SNNs. First, we analyzed the surrogate gradient estimator in adversarial attacks and showed it plays a critical role in achieving a high attack success rate for both BP and Transfer SNNs.

Second, we used the single best gradient estimator to create adversarial examples with different SNN models to measure their transferability with respect to state-of-the-art architectures like Visions Transformers and Big Transfer CNNs. We showed that SNN single-model adversarial examples do not transfer often and there exist multiple SNN/ViT and SNN/CNN pairings that do not share the same set of vulnerabilities to traditional adversarial machine learning attacks.

Lastly, we developed a new attack, MDSE which achieves a high attack success rate against both SNNs and non-SNN models (ViTs and CNNs) simultaneously. MDSE improves attack effectiveness by 91.4% on SNN/ViT ensembles and triples attack performance on adversarially trained SNN ensembles (compared to Auto-PGD). Overall, our comprehensive experiments, analyses and new attack significantly advance the field of SNN security.

## Impact Statements

This paper presents work whose goal is to advance the field of Machine Learning, specifically within the field of Spiking Neural Networks and adversarial Machine Learning. There are many potential societal consequences of our work, however, at the current time we are not aware of any Spiking Neural Networks deployed in applications where adversarial attacks would represent direct harm. The purpose of our work is to advance the field of adversarial machine learning in such a manner that attention is drawn to the issue of adversarial example generation. In this way, future harm may be mitigated through proper security techniques against adversarial manipulation.

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

# A SNN Energy Efficiency

Table 17: ANN and SNN energy consumption.

| Architecture | Dataset | Normalized ANN #OP | Normalized SNN #OP | ANN/SNN Energy |
|---|---|---|---|---|
| SEW-ResNet | CIFAR 10 | 1 | 0.4052 | 12.61 |
| SEW-ResNet | CIFAR 100 | 1 | 0.5788 | 8.83 |
| SEW-ResNet | ImageNet | 1 | 0.5396 | 9.47 |
| Vanilla Spiking ResNet | ImageNet | 1 | 0.6776 | 7.54 |
| Transfer Spiking VGG 16 | ImageNet | 1 | 2.868 | 1.78 |

Benefiting from the binary spikes, the expensive multiplication in DNNs can be greatly eliminated in SNNs. We followed the methodology in Rathi & Roy (2021) and energy model in Rathi & Roy (2021); Horowitz (2014) to theoretically analyze the energy efficiency of SNNs used in this work. For each 32-bit Multiply-Accumulate Operation (MAC) in ANN, energy cost is $4.6pJ$ Horowitz (2014). One MAC of ANN is equivalent to multiple Addition-Accumulation Operations (AAC) of SNN in a time window $T$, number of AAC is calculated as $\#OP_{SNN}=SpikeRate \times T$. Each AAC takes $0.9pJ$ energy. Theoretical comparison is shown in Table 17. ANNs consume 1.78-12.61 times more energy than SNNs. Note that the actual energy efficiency is technology and implementation dependent, and this theoretical calculation is pessimistic: other factors such as data movement, architectural design, etc., which also contribute to neuromorphic chips energy efficiency, are not taken into account. As mentioned in Section 1, various works have reported $10 \times - 276 \times$ energy efficiency over CPU/GPU with dedicated off-the-shelf neuromorphic chips.

## A.1 Fast Gradient Sign Method (FGSM)

The Fast Gradient Sign Method (FGSM) Goodfellow et al. (2015) is a white-box attack that generates adversarial examples by adding noise to the clean image in the direction of the gradients of the loss function:

$$x_{\mathrm{adv}} = x + \epsilon \cdot \mathrm{sign}(\nabla_x \mathcal{L}(x, y; \theta)) \tag{27}$$

where $x_{\mathrm{adv}}$ is the adversarial example, $x$ is the original input, $\epsilon$ is the attack step size parameter, $\mathcal{L}$ is the loss function of the targeted model, $y$ is the true label, $\theta$ represents the model parameters, and $\nabla_x$ denotes the gradient with respect to the input $x$. The attack performs only a single step of perturbation, and applies noise in the direction of the sign of the gradient of the model's loss function.

## A.2 Projected Gradient Descent

The Projected Gradient Descent attack (PGD) Madry et al. (2018) is a modified version of the FGSM attack that implements multiple attack steps. The attack attempts to find the minimum perturbation, bounded by $\epsilon$, which maximizes the model's loss for a particular input, $x$. The attack begins by generating a random perturbation on a ball centered at x and with radius $\epsilon$. Adding this noise to $x$ gives the initial adversarial input, $x_0$. From here the attack begins an iterative process that runs for $k$ steps. During the $i^{th}$ attack step the perturbed image, $x_i$, is updated as follows:

$$x_i = P(x_{i-1} + \alpha \cdot sign(\Delta_x L(x_{i-1}, y; \theta))) \tag{28}$$

where $P$ is a function that projects the adversarial input back onto the $\epsilon$-ball in the case where it steps outside the bounds of the ball and $\alpha$ is the attack step size. The bounds of the ball are defined by the $l_p$ norm.

## A.3 Momentum Iterative Method

The Momentum Iterative Method (MIM) Dong et al. (2018) applies momentum techniques seen in machine learning training to the domain of adversarial machine learning. Similar to those learning methods, the MIM

attack's momentum allows it to overcome local minima and maxima. The attack's main formulation is similar to the formulation seen in the PGD attack. Each attack iteration is calculated as follows:

$$x_i = clip_{x,epsilon}(x_{i-1} + \frac{\epsilon}{t} \cdot sign(g_i)) \tag{29}$$

where $x_i$ represents the adversarial input at iteration $i$, $\epsilon$ is the total attack magnitude, and $t$ is the total number of attack iterations. $g_i$ represents the accumulated gradient at step $i$ and is calculated as follows:

$$g_i = \mu \cdot g_{i-1} + \frac{\Delta_x L(x_{i-1}, y; \theta)}{||\Delta_x L(x_{i-1}, y; \theta)||_1} \tag{30}$$

where $\mu$ represents a momentum decay factor. Due to its similarity of formulation, the MIM attack degenerates to an iterative form of FGSM as $\mu$ approaches 0.

### A.4 Auto-PGD

The Auto-PGD Croce & Hein (2020) is a budget-aware, step size-free variant of PGD. The algorithm partitions the available $N_{\text{iter}}$ iterations to first search for a good initial point. Then, in the exploitation phase, it progressively reduces the step size to maximize the attack results. However, the reduction in step size is not predetermined but is governed by the optimization trend: if the target value grows sufficiently fast, then the step size is likely appropriate; otherwise, it is reasonable to reduce the step size. The gradient update of Auto-PGD follows closely the classic algorithm and adds a momentum term. Let $\eta_i$ be the step size at iteration $i$, then the update step is as follows:

$$z_{i+1} = P\left(x_i + \eta_i \nabla f(x_i)\right)$$
$$x_{i+1} = P\left(x_i + \alpha \cdot (z_{i+1} - x_i) + (1 - \alpha) \cdot (x_i - x_{i-1})\right) \tag{31}$$

where $\alpha \in [0, 1]$ regulates the influence of the previous update on the current one, $z_i$ is the intermediate perturbed point, and $P$ is the projection function.

## B SNN Transferability Study Supplementary Material

Table 18: Full transferability results for CIFAR-10. The first column in each table represents the model used to generate the adversarial examples, $C_i$. The top row in each table represents the model used to evaluate the adversarial examples, $C_j$. Each entry represents $T_{i,j}$ (the transferability) computed using Equation 20 with $C_i$, $C_j$ and either FGSM, PGD, MIM or APGD. For each attack the maximum perturbation bounds is $\epsilon = 0.031$. Based on these results we take the maximum transferability across all attacks and report the result in Table 9. We also visually show the maximum transferability $t_{i,j}$ in Figure 4.

| FGSM | | | | | | | | | | | |
|---|---|---|---|---|---|---|---|---|---|---|---|
| | S-R-BP | S-V-BP | S-V-T5 | S-V-T10 | S-R-T5 | S-R-T10 | VB32 | VB16 | VL16 | C-V | C-R | R101x3 |
| S-R-BP | 78.90% | 15.60% | 12.60% | 13.50% | 18.90% | 16.30% | 6.90% | 5.50% | 4.00% | 13.20% | 16.50% | 4.30% |
| S-V-BP | 14.40% | 64.10% | 31.70% | 31.60% | 34.80% | 36.50% | 6.30% | 6.20% | 5.20% | 28.80% | 35.90% | 6.90% |
| S-V-T5 | 14.20% | 36.40% | 49.70% | 72.40% | 45.70% | 47.60% | 8.00% | 7.90% | 6.50% | 58.70% | 42.70% | 11.60% |
| S-V-T10 | 13.60% | 35.80% | 73.40% | 51.20% | 44.10% | 45.50% | 8.10% | 9.00% | 6.30% | 58.20% | 43.90% | 10.60% |
| S-R-T5 | 9.40% | 16.60% | 17.50% | 18.20% | 24.40% | 34.30% | 4.10% | 4.40% | 2.80% | 19.30% | 30.10% | 4.50% |
| S-R-T10 | 11.40% | 26.00% | 28.60% | 28.70% | 54.50% | 39.80% | 6.00% | 6.90% | 5.00% | 29.70% | 45.90% | 8.20% |
| VB32 | 9.90% | 13.20% | 15.30% | 13.50% | 21.90% | 20.50% | 62.40% | 43.80% | 37.30% | 12.90% | 20.30% | 29.40% |
| VB16 | 8.90% | 11.90% | 10.70% | 10.70% | 18.90% | 17.40% | 30.40% | 60.60% | 43.10% | 10.60% | 16.80% | 25.30% |
| VL16 | 8.10% | 10.00% | 10.70% | 10.30% | 16.30% | 16.70% | 24.40% | 38.40% | 43.50% | 9.90% | 15.20% | 19.20% |
| C-V | 13.60% | 47.60% | 76.50% | 80.20% | 57.90% | 57.70% | 10.60% | 11.90% | 8.30% | 58.80% | 60.70% | 12.60% |
| C-R | 14.70% | 50.00% | 51.90% | 53.60% | 77.80% | 66.10% | 11.60% | 14.70% | 10.00% | 52.40% | 81.40% | 15.90% |
| R101x3 | 8.50% | 7.30% | 7.10% | 7.30% | 11.80% | 9.80% | 3.20% | 5.50% | 3.30% | 6.10% | 9.70% | 13.90% |

| PGD | | | | | | | | | | | |
|---|---|---|---|---|---|---|---|---|---|---|---|
| | S-R-BP | S-V-BP | S-V-T5 | S-V-T10 | S-R-T5 | S-R-T10 | VB32 | VB16 | VL16 | C-V | C-R | R101x3 |
| S-R-BP | 57.10% | 14.80% | 12.10% | 12.20% | 17.80% | 14.50% | 4.80% | 3.20% | 3.10% | 13.30% | 14.90% | 3.00% |
| S-V-BP | 10.90% | 89.90% | 31.30% | 32.40% | 38.60% | 37.70% | 4.60% | 4.00% | 2.60% | 30.40% | 39.00% | 6.00% |
| S-V-T5 | 9.20% | 34.90% | 52.50% | 85.20% | 46.00% | 48.60% | 3.90% | 3.50% | 1.80% | 67.80% | 47.60% | 6.90% |
| S-V-T10 | 10.60% | 34.00% | 92.30% | 52.00% | 45.20% | 45.70% | 4.40% | 3.60% | 2.30% | 66.70% | 45.40% | 7.00% |
| S-R-T5 | 7.00% | 11.40% | 13.00% | 12.60% | 20.90% | 48.20% | 1.30% | 1.30% | 0.80% | 13.30% | 26.10% | 2.20% |
| S-R-T10 | 9.00% | 23.80% | 30.30% | 32.10% | 85.90% | 51.20% | 2.50% | 3.00% | 1.80% | 28.20% | 60.00% | 5.50% |
| VB32 | 7.30% | 6.60% | 5.00% | 4.80% | 8.80% | 6.90% | 97.60% | 63.20% | 39.30% | 4.50% | 5.30% | 32.00% |
| VB16 | 5.80% | 4.20% | 2.70% | 2.40% | 5.60% | 4.70% | 14.80% | 99.80% | 56.80% | 2.10% | 2.90% | 16.90% |
| VL16 | 5.90% | 4.80% | 3.70% | 2.80% | 6.80% | 4.90% | 20.40% | 78.10% | 92.40% | 2.30% | 3.20% | 21.80% |
| C-V | 11.50% | 55.10% | 94.40% | 95.40% | 70.80% | 70.10% | 7.70% | 10.40% | 6.30% | 72.50% | 72.80% | 15.40% |
| C-R | 11.80% | 60.50% | 64.60% | 67.20% | 97.10% | 98.40% | 11.00% | 13.20% | 8.10% | 66.50% | 89.60% | 22.90% |
| R101x3 | 6.00% | 4.40% | 2.60% | 1.60% | 5.50% | 2.50% | 1.20% | 2.70% | 0.90% | 2.00% | 1.90% | 100.00% |

| APGD | | | | | | | | | | | |
|---|---|---|---|---|---|---|---|---|---|---|---|
| | S-R-BP | S-V-BP | S-V-T5 | S-V-T10 | S-R-T5 | S-R-T10 | VB32 | VB16 | VL16 | C-V | C-R | R101x3 |
| S-R-BP | 67.50% | 19.20% | 18.30% | 17.10% | 21.10% | 18.00% | 8.70% | 5.60% | 4.80% | 19.60% | 20.10% | 5.00% |
| S-V-BP | 10.50% | 63.60% | 36.70% | 36.70% | 43.50% | 42.80% | 6.50% | 6.80% | 4.40% | 37.10% | 47.50% | 8.00% |
| S-V-T5 | 9.70% | 38.30% | 59.40% | 96.80% | 54.90% | 55.80% | 3.50% | 4.30% | 2.30% | 76.10% | 52.60% | 7.80% |
| S-V-T10 | 10.80% | 35.00% | 98.00% | 54.90% | 52.00% | 51.30% | 3.90% | 4.10% | 2.50% | 73.70% | 51.50% | 8.40% |
| S-R-T5 | 8.80% | 25.50% | 29.70% | 29.50% | 48.70% | 85.30% | 3.00% | 3.50% | 2.20% | 28.60% | 57.50% | 6.60% |
| S-R-T10 | 10.50% | 36.40% | 43.30% | 43.90% | 97.80% | 68.40% | 5.10% | 5.80% | 3.30% | 38.90% | 79.30% | 9.80% |
| VB32 | 8.40% | 8.20% | 15.60% | 15.50% | 21.60% | 16.50% | 100.00% | 70.50% | 47.80% | 6.10% | 9.40% | 40.40% |
| VB16 | 6.30% | 6.80% | 11.70% | 11.50% | 16.20% | 11.80% | 22.90% | 100.00% | 71.40% | 3.80% | 7.40% | 25.50% |
| VL16 | 6.50% | 6.10% | 13.40% | 14.10% | 19.60% | 13.20% | 26.50% | 87.00% | 99.00% | 5.80% | 7.50% | 26.90% |
| C-V | 11.60% | 65.40% | 98.10% | 98.60% | 78.80% | 82.50% | 13.80% | 14.30% | 10.20% | 83.90% | 83.10% | 21.50% |
| C-R | 11.10% | 66.10% | 69.30% | 71.90% | 98.30% | 99.10% | 15.20% | 17.20% | 12.10% | 82.20% | 97.80% | 29.30% |
| R101x3 | 7.70% | 5.30% | 6.80% | 7.50% | 11.40% | 9.60% | 3.30% | 7.50% | 3.30% | 2.40% | 4.10% | 100.00% |

| MIM | | | | | | | | | | | |
|---|---|---|---|---|---|---|---|---|---|---|---|
| | S-R-BP | S-V-BP | S-V-T5 | S-V-T10 | S-R-T5 | S-R-T10 | VB32 | VB16 | VL16 | C-V | C-R | R101x3 |
| S-R-BP | 92.00% | 19.30% | 16.60% | 15.40% | 20.20% | 16.20% | 6.80% | 5.10% | 4.80% | 15.50% | 18.60% | 4.30% |
| S-V-BP | 15.30% | 88.60% | 46.20% | 46.60% | 51.80% | 51.50% | 10.10% | 9.80% | 6.50% | 44.00% | 52.30% | 12.20% |
| S-V-T5 | 12.10% | 45.10% | 60.10% | 80.90% | 54.10% | 55.50% | 8.70% | 9.20% | 5.50% | 67.80% | 53.40% | 13.30% |
| S-V-T10 | 13.00% | 42.40% | 89.10% | 57.60% | 52.90% | 52.30% | 8.50% | 9.10% | 5.70% | 68.30% | 51.10% | 12.10% |
| S-R-T5 | 10.10% | 23.10% | 27.70% | 27.80% | 38.70% | 66.40% | 3.70% | 4.40% | 3.70% | 26.60% | 47.50% | 4.80% |
| S-R-T10 | 11.70% | 38.80% | 47.10% | 48.90% | 88.20% | 64.00% | 8.80% | 8.50% | 6.40% | 41.60% | 72.90% | 12.80% |
| VB32 | 10.70% | 14.40% | 13.30% | 12.20% | 20.90% | 18.40% | 95.90% | 83.70% | 75.10% | 13.00% | 18.20% | 60.40% |
| VB16 | 7.50% | 9.70% | 11.10% | 10.00% | 14.40% | 13.90% | 57.40% | 99.40% | 88.90% | 9.40% | 14.30% | 42.90% |
| VL16 | 7.90% | 9.70% | 9.10% | 9.10% | 14.80% | 13.70% | 55.30% | 78.40% | 91.60% | 8.60% | 13.20% | 44.20% |
| C-V | 14.40% | 63.50% | 94.50% | 95.70% | 73.40% | 76.30% | 12.80% | 14.90% | 10.90% | 78.40% | 77.70% | 19.40% |
| C-R | 15.20% | 67.20% | 74.60% | 74.00% | 96.10% | 89.20% | 15.40% | 20.00% | 13.70% | 73.50% | 98.30% | 28.90% |
| R101x3 | 7.50% | 7.20% | 5.70% | 4.90% | 11.80% | 7.70% | 8.60% | 20.00% | 12.30% | 5.60% | 8.20% | 100.00% |

