# OpenReview forum: "Attacking the Spike: On the Security of Spiking Neural Networks to Adversarial Examples"
_TMLR — Rejected by TMLR_

### Review · Reviewer_w6PC · 2024-07-12

**Summary Of Contributions:**

Spiking neural networks (SNNs) offer high energy efficiency and improved classification performance, but their robustness against adversarial attacks remains underexplored. This paper tackles this problem by analyzing the SNNs under adversarial attacks. Firstly, it demonstrates the critical role of surrogate gradient estimation techniques in the success of white-box adversarial attacks on SNNs. Secondly, it analyzes the transferability of adversarial attacks across SNNs, Vision Transformers (ViTs), and CNNs, revealing gaps in exploiting multiple surrogate estimators and cross-model attack effectiveness. Lastly, the study introduces the Mixed Dynamic Spiking Estimation (MDSE) attack, which improves the attack success rates on both SNNs and non-SNN models, showcasing improvements in attack effectiveness by up to 91.4% on SNN/ViT model ensembles.

**Audience:**

Yes

**Broader Impact Concerns:**

The Impact Statements have been discussed by the authors.

**Claims And Evidence:**

Yes

**Requested Changes:**

1. Please discuss more in detail what are the limitations of the related work and how these issues are addressed in the proposed method.

2. Please discuss the proposed attack algorithm more in detail, describing the design decisions made to develop it.

3. Please describe in more detail the experimental setup and tool flow used to conduct the experiments.

4. In Sections 3, 4, and 6, each figure and table should have lists of observation points to discuss the observations derived from the results.

5. The results have only be conducted on static datasets. Please make experiments on event-based datasets, which are more suitable for SNNs.

**Strengths And Weaknesses:**

Strengths:

1. The work is relevant to the community.

2. The results have shown the effectiveness of the proposed method.

Weaknesses:

1. Some parts of the paper require better clarity.

2. The main technical section describing in detail the technical contributions is missing.

---

> ### Author Response · Authors · 2024-09-13
>
> We thank reviewer w6PC for their insightful comments. Much of what they are asking is already contained within the paper. In our second draft we will do our best to make edits to make these parts more apparent. Below we address each of the reviewer's comments individually.
>
> **1. Please discuss more in detail what are the limitations of the related work and how these issues are addressed in the proposed method.**
>
> For each of our contributions we specifically address where the literature is lacking and what our majority novelty and contribution is. Specifically, we address how we are different from existing attacks (A) and how we are unique in terms of adversarial training (B).
>
> (A) For existing attack methodologies we specifically mention the limitations of existing work in section 5.2:
>
> "There are several pertinent advantages of the MDSE attack over other white-box attacks. The Self-Attention Gradient Attack (SAGA) was proposed in~\cite{mahmood2021robustness} for attacking multiple models, similar to MDSE. However, in regards to non-SNN models, SAGA has two key limitations that MDSE overcomes. Assume a model ensemble containing the set of models ${E = S \cup V }$ and ${|E|=M}$. Every model $m$ requires its own weighting factor such that ${ \overrightarrow{\alpha}=({\alpha_{1},\cdots,\alpha_{m},...,\alpha_{M}}) }$. If these hyperparameters are not properly chosen, the attack performance of SAGA degrades significantly. This was demonstrated in Mahmood et al. In MDSE, these coefficients are adaptively updated at every iteration of the attack, removing this pitfall. The second drawback of SAGA is that once ${\overrightarrow{\alpha}}$ is chosen for the attack, it is fixed for every sample and for every iteration of the attack. This makes choosing ${  \overrightarrow{\alpha}}$ incredibly challenging as each scalar hyperparameter $\alpha_{m}$ must either perform well for the majority of samples or have to be manually selected on a per sample basis (since $\alpha_{m} \in \mathbb{R}$). In MDSE, ${\overrightarrow{\alpha}}$ is fine grained on a per sample and per pixel basis i.e., $\alpha_{m} \in \mathbb{R}^{b \times h \times w \times c}$ where $b \times h \times w \times c$ represent the batch size, height, width and number of color channels for the input to MDSE. In addition to overcoming two key limitations of SAGA for non-SNN models, MDSE also leverages the dynamic gradient estimation scheme. This makes MDSE stronger for ensembles that include SNNs, something SAGA lacks."
>
> (B) For adversarial training we discuss the limitations of the current literature and why we want to extend it to SNN's in section 2.2:
>
> "To the best of our knowledge, we are the first to implement DM and FAT adversarial training techniques on SNNs. We are also the first to compare existing SNN-specific adversarial training (TIC and HIRE) to DM and FAT. While this alone is not a major contribution, in the context of developing adversarial attacks, it is critical to include these types of analyses. This is because existing adversarial attacks can readily be adapted to new undefended architectures yielding a high attack success rate. However, on defended models or model ensembles, existing adversarial attacks may not be effective. Experimenting with adversarial training defense methods are key for accurately assessing the robustness of SNNs to SOTA adversarial attacks."
>
>
> **2. Please discuss the proposed attack algorithm more in detail, describing the design decisions made to develop it.**
>
> In section 5 we specifically address this issue before we give details related to our attack:
>
> "We have demonstrated two major issues currently facing white-box adversarial attacks on SNNs. First, in Section 3, we showed the choice of surrogate gradient estimator heavily influences how successful the attack is. While certain estimator functions performed better than others, the result was highly dependent on the dataset and model. There was no single estimator that worked best across all models and all datasets. The second major issue is that even when an effective gradient estimator is employed, adversarial examples created by attacks like Auto-PGD are not misclassified by SNNs and non-SNN models simultaneously. We demonstrated this result in Section 4. State-of-the-art white-box attacks do not transfer well and do not exploit the capabilities of different gradient estimators for SNNs. To address the two major problems, we propose a new white-box attack, the Mixed Dynamic Spiking Estimation (MDSE) Attack. Our new attack is comprised of two main components, dynamic estimation of the spiking gradients and mixing of the gradients when attacking multiple models. The pseudo-code for the MDSE attack is given in Algorithm…"
>
> For giving details for our attack algorithm, a full pseudo-code is given in Algorithm 1. The mathematics for our attack are fully explained and justified in sections 5.1 and 5.2.

---

> ### Author Response · Authors · 2024-09-13
>
> **3. Please describe in more detail the experimental setup and tool flow used to conduct the experiments.**
>
> All experiments are conducted in PyTorch and spikingjelly using a workstation equipped with an AMD Ryzen Threadripper PRO 3975WX 32-core processor, 256 GB of memory, and two NVIDIA GeForce RTX 3090Ti GPUs. The experimental setup is thoroughly detailed in Section 3.2, Section 4.1, and Section 6.1. In the revised paper, we have extended these details for further clarity. Additionally, we will provide a link to our open-source code on GitHub once the paper is published.
>
>
> **4. In Sections 3, 4, and 6, each figure and table should have lists of observation points to discuss the observations derived from the results.**
>
> Thank you for your insightful feedback. In Section 3.2, Tables 1, 2, 3, and 4 provide the detailed results corresponding to Figure 2. We have outlined various observations and discussed the results for CIFAR-10, CIFAR-100, and ImageNet in the "Vanilla SNN Experimental Analysis" section.
>
> The key takeaways from Section 4 are clearly stated in Section 4.1, where we have enumerated the main observations under #1 and #2. Section 4 as a whole focuses on transferability, demonstrating that no single white-box attack achieves high success rates across both SNN and non-SNN models simultaneously. Additionally, we observe a similar lack of transferability between different types of SNN models. This highlights the need for an effective multi-model SNN white-box attack, which we subsequently develop in the form of MDSE.
>
> In Section 6, the primary objective is to show that our MDSE attack outperforms several other state-of-the-art (SOTA) attacks, including Auto-PGD, MIM, and SAGA. To clearly highlight this, we have bolded the results in each table in Section 6 where MDSE surpasses other attacks.
>
> We appreciate the reviewer’s suggestion to improve the readability of tables and figures by providing clearer takeaways, and we will update them accordingly. In the revised version, we will also expand on the observation points to provide more detailed discussions, as recommended.
>
> **5. The results have only be conducted on static datasets. Please make experiments on event-based datasets, which are more suitable for SNNs.**
>
> It is important to note that our paper focuses on image classification tasks for SNNs. This has no direct relation to event-based datasets aside from the fact that SNNs have been used in both fields. From a technical perspective, we will later explain why our proposed attack is not directly applicable. However, the first thing that must be mentioned is that in general, the literature considers SNN papers that only work in the adversarial image domain (without event-datasets) significant contributions worthy of publication. Here we mention just a few of the many SNN papers that only considered image datasets in the context of security (and were published at top venues):
>
> - [A] Kim, Y., et al. (2023). *Exploring temporal information dynamics in spiking neural networks.* AAAI Conference on Artificial Intelligence, 37(7), 8308-8316.
> - [B] Kundu, S., et al. (2021). *Hire-snn: Harnessing the inherent robustness of energy-efficient deep spiking neural networks by training with crafted input noise.* IEEE/CVF International Conference on Computer Vision, 5209-5218.
> - [C] Kim, Y., et al. (2022). *Privatesnn: privacy-preserving spiking neural networks.* AAAI Conference on Artificial Intelligence, 36(1), 1192-1200.
>
>
> Technical Answer: There are fundamental differences between image data and event dataset. Image data is defined in real domain R. However, the events in the event-dataset are defined in {0, 1}, i.e., the value can only be 0 or 1. The attack method (MDSE) developed in this work generates adversarial examples by calculating a perturbation and adding it to the original image. Note that the perturbation is also defined in real domain R, which is incompatible with the binary discrete event which is defined in {0, 1}. This data format incompatibility is discussed in (Liang et al., 2021):  https://arxiv.org/abs/2001.01587. It may be possible in the future work to have MDSE tested on event-datasets, but this would come only after significant modifications to the original algorithm. This could be an exciting future work, but it is simply not applicable to our current paper which focuses on the image domain.

---

### Review · Reviewer_X25k · 2024-07-15

**Summary Of Contributions:**

This paper focuses on understanding the adversarial attack of SNNs. Specifically, authors discover that successful white-box adversarial attacks on SNNs are highly dependent on the underlying surrogate gradient estimation technique. Authors use the best single surrogate gradient estimation technique to analyze the transferability of adversarial attacks on SNNs. Moreover, authors develop a new attack, which utilizes a dynamic gradient estimation scheme to fully exploit multiple surrogate gradient estimator functions.

**Audience:**

No

**Broader Impact Concerns:**

There do not exist broader impact concerns.

**Claims And Evidence:**

Yes

**Requested Changes:**

1. The training process of SNNs in Section 2.1 will be more rigorous if such a training process is represented in a mathematical form like SGD of CNNs.

2. Authors do not clarify how Eq. (2) ensures "that the Fisher information shows a similar trend for all timesteps." It will be more clear if authors can clarify how Fisher information is represented/defined in SNNs, and then explains how to control the Fisher Information.

3. For Section 2.2, authors should first classify the adversarial training methods of SNNs into several kinds, and then from each kind choose one method to introduce. Now, although the reason for selecting each method is given, it is hard to say these methods are representative.

4. How about l2 attack? In this paper, authors seem only consider l∞ attack. It will be better authors conduct some experiments based on l2 attacks to verify the proposed conclusions.

5. Experimental results will be more convincing if authors conduct more experiments on other datasets and more network architectures.

**Strengths And Weaknesses:**

The topic authors choosing is interesting.

---

> ### Author Response · Authors · 2024-09-13
>
> **1. The training process of SNNs in Section 2.1 will be more rigorous if such a training process is represented in a mathematical form like SGD of CNNs.**
>
> Thank you for your valuable feedback. In response, we have added further details in Section 2.1 of the revised version
>
> **2. Authors do not clarify how Eq. (2) ensures "that the Fisher information shows a similar trend for all timesteps." It will be more clear if authors can clarify how Fisher information is represented/defined in SNNs, and then explains how to control the Fisher Information.**
>
> The Fisher Information Matrix (FIM) quantifies the amount of information inside a model obtained from a given data when the model parameters are perturbed (Fisher 1925). In paper [1], they use the following formula for quantifying the amount of Fisher information across different timesteps:
> $
> I_t(\theta) = \frac{1}{N} \sum_{n=1}^{N} \left\| \nabla_{\theta} \log f_{\theta}(y | x_{i \leq t}^n) \right\|^2
> $.
> According to Definition 1 in [1], the log posterior $\( \log f_{\theta}(y | x_{i \leq t}) \)$ can be represented as a loss function $\( L_t(\theta) \)$, such as cross-entropy loss, where the final layer’s outputs are accumulated at timesteps before they are converted to probabilities (e.g., with a softmax layer). Thus, the Fisher information $\( I_t(\theta) \)$ can be represented as:
> $I_t(\theta) = \mathbb{E}\left[ \left\| \nabla L_t(\theta) \right\|^2 \right]$.
> A decreasing loss $ L_t(\theta) $ leads to Fisher information degradation because the gradient $ \nabla L_t(\theta)$ decreases as the loss converges to a local minimum. On the other hand, if we disturb the model to converge a loss value, Fisher information cannot become smaller.
> Based on this hypothesis, the authors control the Fisher information value at each timestep by manipulating the loss function during training. Specifically, they force the loss function to have a value around $ \alpha $, as shown in $L_t(\theta, \alpha) = |L_t(\theta) - \alpha|$. The equation is applied across $ T $ timesteps to ensure that the Fisher information follows a consistent trend across all time points, as further represented in $L(\theta, \alpha) = \frac{1}{T} \sum_{t=1}^{T} L_t(\theta, \alpha)$. By adjusting the value of $\alpha$, the authors can approximately control the relative magnitude of the Fisher information across the timesteps.
>
> For full details, please refer to the reference paper [1].
>
>
>
> **3. For Section 2.2, authors should first classify the adversarial training methods of SNNs into several kinds, and then from each kind choose one method to introduce. Now, although the reason for selecting each method is given, it is hard to say these methods are representative.**
>
> In our paper, we choose four different state-of-the-art defensive training methods to evaluate the effectiveness of our proposed adversarial attack. TIC is designed to improve the robustness of training-based SNN, and HIRE-SNN is an adversarial training for conversion-based SNN. FAT and DM are two well-performed adversarial training methods for DNN and we adapted them in SNN training for evaluation. The primary objective of this paper is to investigate the vulnerabilities of SNNs and the transferability of attacks across different architectures, including SNNs, ViTs, and CNNs. We selected these methods to demonstrate that, despite robust adversarial training, SNNs remain susceptible to adversarial attacks, highlighting their vulnerability.
>
>
> ---
>
> **References:**
>
> [1] Kim, et al. Exploring temporal information dynamics in spiking neural networks. AAAI 2023.

---

> ### Author Response · Authors · 2024-09-13
>
> **4. How about l2 attack? In this paper, authors seem only consider l∞ attack. It will be better authors conduct some experiments based on l2 attacks to verify the proposed conclusions.**
>
> In general the l-infinity attack is the most widely accepted attack and many top tier papers limit their focus to only this threat model. For example:
>
> https://arxiv.org/abs/2302.12252 (ICLR 2023)
>
> https://arxiv.org/pdf/2207.03574 (ICML 2022)
>
> https://arxiv.org/pdf/2002.11242 (ICML 2020)
>
> Just to name a few off hand. We agree with the reviewer that the l2 norm is an interesting experimental setup but it is simply beyond the scope of the paper as this would require retraining ALL the adversarial defenses to account for this norm. This is due to the fact that defending against one type of norm attack is not sufficient to defend against other types of norm attacks (see: https://arxiv.org/pdf/1909.04068 for support for this claim). Therefore, we leave extending our attack to other norms (and extending SNN adversarial training to account for multiple norms) as an exciting possible future work.
>
>
> **5. Experimental results will be more convincing if authors conduct more experiments on other datasets and more network architectures.**
>
> We agree that expanding our research to other tasks is of significant importance. It's worth noting that various tasks, such as text and audio, may exhibit distinct properties and require different model structures. Our paper has undertaken a comprehensive study on the image domain, encompassing 3 image datasets, 4 different white-box attacks, 19 classifier models (7 for CIFAR-10, 7 for CIFAR-100, and 5 for ImageNet), and 4 different adversarial training methods (2 SOTA SNN based training and adapt 2 well-performed adversarial training methods for DNN).
>
> In addition to the existing results, we have included two new SNNs—VGG-11 (SR*-V11) and WideResNet (SR*-W16) with a depth of 16 and width of 4 for CIFAR-10—provided by the state-of-the-art SNN adversarial training method [2]. The evaluation results for these models are included in Section 6.2.4 of the revised version.  Furthermore, we have expanded the discussion on the proposed attack by including an ablation study, which can be found in Section 6.2.5.
>
> Exploring new domains presents an interesting direction for future research.
>
> ---
>
> **References:**
>
> [2] Liu et al. Enhancing Adversarial Robustness in SNNs with Sparse Gradients. ICML 2024.

---

### Review · Reviewer_QwZM · 2024-09-07

**Summary Of Contributions:**

This paper presents two key findings based on extensive experiments on CIFAR-10, CIFAR-100, and ImageNet: (1) surrogate gradients significantly influence the attack success rate on SNNs, and (2) adversarial examples generated using a single surrogate gradient do not transfer well from SNNs to CNNs or ViTs. Building on these observations, the authors propose a new attack method, MDSE, which achieves a high success rate against both SNNs and non-SNN models.

**Audience:**

Yes

**Broader Impact Concerns:**

The authors have addressed the broader impact of their work. I find their statement satisfactory and aligned with the goals of responsible research.

**Claims And Evidence:**

Yes

**Requested Changes:**

As illustrated above, the requested changes are:

(1)Include a discussion of related works and provide comparisons where necessary.

(2)Add ablation studies to demonstrate the effectiveness of ensembling surrogate gradients without ensemble attacked models.

(3)Provide details on the average time required to generate a single adversarial example using MDSE.

(4)Address the minor weaknesses mentioned.

**Strengths And Weaknesses:**

Strength

(1)The concept of dynamically changing surrogate gradients is novel.

(2)The authors conduct extensive experiments, leading to two key insights.

(3)The proposed MDSE method achieves state-of-the-art performance compared to existing attack methods.

Weakness

(1)The paper does not mention or compare with prior works [1,2] which have attempted to attack SNNs using ensemble surrogate gradients. Unlike the proposed MDSE, these methods consider an attack successful if the model is fooled by any of the surrogate gradients for a given input image. This approach differs from the Max Auto/MIM strategy in this work, where the same surrogate gradient is applied to all input samples, and the best one is selected based on the final success rate.

(2)MDSE ensembles both surrogate gradients and attacked models, as shown in Equation (21). Previous works have demonstrated that model ensembling can significantly improve transferability [3]. However, the contribution of ensembling surrogate gradients, which is the key novelty of this paper, requires further validation to prove its effectiveness.

(3)The paper does not provide sufficient information regarding the time complexity of MDSE. It would be valuable to know whether MDSE is time-consuming and how much time, on average, it takes to generate a single adversarial example.

Other minor weaknesses：

(4)There is redundancy in presenting the same results in both tables and figures. For instance, Figure 2 uses a line chart to illustrate the performance of vanilla SNNs with different surrogate gradients, while Tables 1-4 provide the numerical results for the same experiment.

(5)The y-axis label in Figure 2 is incorrect. It is currently labeled as "accuracy," but it should represent the "attack success rate."

Reference:

[1] Ozdenizci et al. Adversarially robust spiking neural networks through conversion. arXiv 2023.

[2] Liu et al. Enhancing Adversarial Robustness in SNNs with Sparse Gradients. ICML 2024.

[3] Dong et al. Boosting Adversarial Attacks with Momentum. CVPR 2018.

---

> ### Author Response · Authors · 2024-09-13
> **(1) Include a discussion of related works and provide comparisons where necessary.**
>
> The reviewer references several papers where an ensemble attack approach is used. It is important to understand that HOW they measure attack success rate is fundamentally different from the metric used in our paper. We explain it here for convenience and also include this updated explanation in the revised version of our paper (Section 6.1):
>
> **Ensemble Attack Success Rate** In the context of proposing a new attack, it is important to define what constitutes a "successful" attack so that the attack success rate can be measured, and different white-box attacks can be directly compared. It is important to note that in the literature there are two established methodologies for measuring the attack success rate on model ensembles. We denote each of these methodologies as *any* and *all*. We will first mathematically define these and then justify our choice of attack measurement.
>
>
> When attacking an ensemble of models we can consider this group as $\Delta = S \cup V$ where $S$ is the set of SNN models and $V$ is the set of non-SNN models. An adversarial example $x_{adv}$ with corresponding clean class label $y$ is considered a successful adversarial attack under the *any* attack metric if the following condition holds: $\exists c \in \Delta, \text{s.t.} c(x_{adv}) \neq y$. Essentially this means under the *any* metric, as long as the adversarial example is misclassified by any of the models in the ensemble the attack is considered successful. This metric has been used in previous literature including [1], [2]. The *all* metric defines a successful adversarial attack as follows: $\forall c \in \Delta, c(x_{adv}) \neq y$. Under this metric an adversarial example is only considered a successful attack if all models in the ensemble produce the wrong class label. This metric has been adopted in many works including [3], [4].
>
> The *all* metric has the following three advantages. First, this metric accurately reflects attack success rate even when majority voting is used in ensembles. Under the *all* condition, even if the defender attempts to form a consensus from model voting, no class label is correct so the defender will never be able to predict the correct class label from the ensemble outputs. Second, the *all* metric reflects the worst case scenario for the attacker, which presents a more realistic lower bound on the attack performance. If even one model in the ensemble has $c(x_{adv})=y$ then the assumption of this metric is that the defender picks that model and deduces the correct class. It is a standard practice in security to assume the worst case for the defender (when proposing a new defense) [5] and the worst case for the attacker (when proposing a new attack). Assuming a stronger defender gives a lower bound on the performance of the attack. Third and lastly, the *any* metric suffers from an issue which we denote as the "weakest model link" that the *all* metric does not suffer from. Assume a set model ensemble $S$ contains $n$ models where $n \geq 2$. Let us denote a weak model $c_{w}$ where $c_{w} \in \Delta$ and $\forall x_{adv} \in X, c_{w}(x_{adv}) \neq y$ where $X$ is the set of all adversarial examples on which we wish to measure the attack success rate. Further consider the case where one or more models in set $S$ are not successfully attacked, $\forall x_{adv} \in X, \exists c \in \Delta, \text{s.t.} \{c(x_{adv})=y\}$. Under the *any* metric, the attack success rate would be reported as $100\%$, even though at least one model in the ensemble correctly identified $x_{adv}$. In reality, the defender still has probability $p \geq \frac{1}{n}$ of picking the right class label if randomly selecting between ensemble classifiers, when classifiers do not all return the same class label. In short, by adding a weak model $c_{w}$ to any ensemble, the attack success rate can be artificially boosted if the *any* metric is used.  For all of these reasons we use the *all* metric when measuring attack success rate for each attack in our analyses in the experiments.
>
>
>
> [1] Ozdenizci et al. *Adversarially robust spiking neural networks through conversion*. arXiv 2023.
> [2] Liu et al. *Enhancing Adversarial Robustness in SNNs with Sparse Gradients*. ICML 2024.
> [3] Liu et al. *Delving into transferable adversarial examples and black-box attacks*. ICLR 2017.
> [4] Mahmood et al. *On the robustness of vision transformers to adversarial examples*. ICCV 2021.
> [5] Carlini, et al. *On evaluating adversarial robustness*. arXiv 2019.

---

> > ### Author Response · Authors · 2024-09-13
> >
> > We would like to thank the reviewer for pointing us toward the reference papers. Based on this feedback, we conducted experiments using pretrained checkpoint models from Reference [2] for evaluation. Although we found the code for Reference [1], time constraints limited our ability to train a model for evaluation.
> >
> > We tested two SNNs provided by Reference [2], denoted as SR*, which implement a sparsity regularization strategy with adversarial training. Specifically, we used the checkpoints for VGG-11 (SR*-V11) and WideResNet (SR*-W16) with a depth of 16 and width of 4 for CIFAR-10. In our evaluation, we paired these models with the DM-R18 SNN to conduct the attacks.
> >
> > For the attack, we used 1,000 correctly classified, class-balanced samples from the CIFAR-10 validation set. The attack allowed a maximum perturbation of \epsilon = 0.031ϵ=0.031 in the l_{\infty}l∞​ norm. We compared the performance of our MDSE attack with that of Auto-PGD, MIM, PGD, and SAGA attacks. The attacks were conducted with a batch size of 50, though this was reduced when GPU memory was insufficient. We executed each attack with various SG configurations and presented the best results.
> >
> > As shown in the table below, our MDSE attack consistently outperforms the other attacks, demonstrating its strength. We have included these results, along with the details of the two SNNs, in Section 6.2.4 of the revised version.
> >
> > | Model 1  | Model 2  | Max MIM | Max PGD | Max Auto | SAGA | MDSE  |
> > |----------|----------|---------|---------|----------|------------|-------|
> > | DM-R18   | SR*-V11  | 16.0%   | 15.5%   | 18.0%    | 18.9%      | 27.6% |
> > | DM-R18   | SR*-W16  | 15.3%   | 15.1%   | 14.7%    | 19.5%      | 25.8% |

---

> ### Author Response · Authors · 2024-09-13
> **(2) Add ablation studies to demonstrate the effectiveness of ensembling surrogate gradients without ensemble attacked models.**
>
> We would like to kindly clarify a point regarding the summary of our attack results. While previous work has shown that ensemble attacks work well, we are further building upon this by selecting the surrogate gradient for each individual model in each step of the attack before combining gradients to create the adversarial noise in each step of the attack.
>
> To highlight the effectiveness of the greedy gradient estimator algorithm, we conducted an ablation study and summarized the results as follows:
> | Number of SGs | AutoPGD | SAGA  | MDSE  |
> |--------|---------|-------|-------|
> | 1 (Arctan)    | 21.7%   | 21.0% | 23.2% |
> | 4             | 20.1%   | 21.0% | 28.4% |
> | 5             | 21.4%   | 21.2% | 27.6% |
> | 7             | 22.2%   | 20.4% | 29.7% |
>
> This table shows the attack success rates for AutoPGD, SAGA and MDSE for adversarial examples generated by DM-R18 and FAT-R18 SNN for CIFAR-10, using different numbers of SGs used during the attack. We use a single SG (Arctan), which is generally the best performance surrogate gradient estimator as studied in Section 3. Then we extend the number of SGs to 4 (Arctan, PWL, Erfc, and Rectangle), 5 (Arctan, PWL, Erfc, Rectangle, and Sigmoid), and 7 (Arctan, PWL, Erfc, Rectangle, Sigmoid, PWE, and Fast Sigmoid) to run the attacks. The results indicate that while adding more SGs offers limited improvements for AutoPGD and SAGA attacks, it significantly enhances the attack effectiveness when using MDSE, with more SGs contributing to stronger attacks.
>
> Additionally, we present another table to further demonstrate the attack success rates for different adversarially trained SNN pairs using our proposed MDSE attack, with both a single SG and multiple SGs:
>
>  | Model 1  | Model 2         | CIFAR10 MDS with 1 SG | CIFAR10 MDSE | CIFAR100 MDS with 1 SG | CIFAR100 MDSE |
> |----------|-----------------|-----------------------|---------------|------------------------|---------------|
> | TIC-R19  | HIRE-V16/V11     | 66.4%                 | 68.5%         | 72.9%                  | 79.3%         |
> | FAT-R18  | HIRE-V16/V11     | 22.6%                 | 47.1%         | 50.1%                  | 54.5%         |
> | DM-R18   | HIRE-V16/V11     | 21.4%                 | 38.5%         | 38.0%                  | 39.0%         |
> | DM-R18   | FAT-R18          | 23.2%                 | 29.7%         | 37.4%                  | 41.0%         |
> | FAT-R18  | TIC-R19          | 23.8%                 | 27.1%         | 46.6%                  | 47.7%         |
> | DM-R18   | TIC-R19          | 22.2%                 | 25.4%         | 35.8%                  | 38.5%         |
>
> This table provides a detailed comparison of the attack success rates for different adversarially trained SNN pairs using our proposed MDSE attack. We observe that even with a fixed SG choice, our mixed multi-model attack, featuring adaptively updated coefficients for each model, surpasses other attacks, as compared to the results presented in Table 14 of the revised paper.
>
> Moreover, the attack success rate consistently improves for all model pairs when dynamic gradient estimation with multiple SG options is employed at each attack step. Specifically, we achieve an improvement of approximately 2.1\% to 24.5\% on CIFAR-10 and 1.0\% to 6.4\% on CIFAR-100, even against these robust adversarially trained SNNs.
>
> We have included these findings and discussions in Section 6.2.5 of the revised paper, along with the corresponding tables.

---

> ### Author Response · Authors · 2024-09-13
> **(3) Provide details on the average time required to generate a single adversarial example using MDSE.**
>
> From a theoretical standpoint both MDSE and any single white-box attack that uses the maximization strategy we outlined in the paper will have at most a p time difference in terms of  asymptotic complexity under the assumption that the number of GPUs (g) is at least equal to the number of models (n). Consider the forward pass in max MIM for n models m_1, m_2,...,m_n. The runtime for a single step of the attack will be O(m_1+m_2+...+m_n) so the largest model (which we can denote as mk) will dominate and hence the overall attack complexity for one step will be O(m_k). Now let us consider MDSE. For a single step each model further needs to evaluate p surrogate gradients hence the complexity will be O(p*(m_1+m_2+...+m_n)). This again reduces to the complexity of the largest model and therefore will be O(p*m_k). However if we consider the case where each surrogate gradient can be computed in parallel then the complexity becomes O(m_k). We can add such an analysis more formally to the appendix of the paper, but we don’t want to write this in the main paper for the following reasons. First, the underlying assumptions on computational complexity depend on the relationship between the number of GPUs and number of models under attack (g=n). Second, the underlying assumption in the analysis is that the largest model m_k is able to fit within the GPU memory and in general each GPU is able to load each model into GPU memory. This analysis changes drastically if models must be partially computed on the CPU or iIf part of the models do not fit into memory, or there are less GPUs than models. In any of these cases the analysis is no longer valid and must be re-derived. In general, the actual runtime for generating adversarial examples in practice using MDSE is in the order of minutes for CIFAR-10 and ImageNet but is highly dependent on the hardware setup. When the paper is published, on our Github page corresponding to the paper, we detail the hardware settings we used and approximate runtimes.
>
> Overall we specifically do not include a time complexity analysis because the results could be construed as misleading. The complexity of the attack in practice has similar running times to SAGA. It is important to note that in the white-box attack setting, the most expensive part of the computation is almost always the backward pass on the model. Hence, most white-box attack works exclude explicit discussions of run time as it is trivial. See the following attack papers which do not include explicit attack time complexity analyses for example:
>
> AutoAttack (APGD): https://arxiv.org/pdf/2003.01690
>
> SAGA: https://arxiv.org/abs/2104.02610#

---

> ### Author Response · Authors · 2024-09-13
> **(4) Address the minor weaknesses mentioned.**
>
> We sincerely thank the reviewer for their valuable feedback, which has greatly helped us improve the content of the paper. As suggested, we have updated the y-axis label of Fig. 2 in the revised version.
>
> Regarding the presentation of the numerical results, our intention is to provide readers with an overview through the figures while also including detailed results for those who may be interested in further specifics. We will carefully reconsider the structure of this content based on your suggestion, and we truly appreciate your insightful advice.

---

### Decision · Action_Editor_Tmoh · 2024-10-16

**Recommendation:** Reject

**Comment:**

The reviewers raised concerns on the considered attacks are representative enough to support the claims of the manuscript. The practical feasibility of the proposed method is unclear. Furthermore, one reviewer noted a potential computational overhead which raised concerns about the practical feasibility of the approach.
The author's claim to show the dependency of successful white-box adversarial attacks on the surrogate gradient estimation technique, and do not mention that other work showed this (Ozdenizci & Legenstein) previously.

**Audience:**

The manuscript is potentially interesting for parts of the TMLR audience.

**Claims And Evidence:**

The claims of the paper are not fully supported by the provided evidence. It is not clear that the chosen adversarial training methods are representative enough to support the claims. The authors did not consider L2-norm attacks. One reviewer argues that such attacks need to be considered to support claims on the effectiveness of the proposed method. He also noted that the claim that the L-infinity attack is the most widely used is not supported. Furthermore, one reviewer raised concerns about the practical feasibility of the approach due to potential computational overhead.
The authors claim: "First, we show that successful white-box adversarial attacks on SNNs are highly dependent on the underlying surrogate gradient estimation technique, even in the case of adversarially trained SNNs." They do not mention that this has been shown in previous work (Ozdenizci & Legenstein).

**Resubmission Of Major Revision:**

The authors may consider submitting a major revision at a later time.